# A DARPin-based molecular toolset to probe gephyrin and inhibitory synapse biology

**Benjamin FN Campbell[1], Antje Dittmann[2], Birgit Dreier[3], Andreas Plückthun[3], Shiva K Tyagarajan[1]***

[1]Institute of Pharmacology and Toxicology, University of Zurich, Zurich, Switzerland; [2]Functional Genomics Centre, University of Zurich, Zurich, Switzerland; [3]Department of Biochemistry, University of Zurich, Zurich, Switzerland

**Abstract** Neuroscience currently requires the use of antibodies to study synaptic proteins, where antibody binding is used as a correlate to define the presence, plasticity, and regulation of synapses. Gephyrin is an inhibitory synaptic scaffolding protein used to mark GABAergic and glycinergic postsynaptic sites. Despite the importance of gephyrin in modulating inhibitory transmission, its study is currently limited by the tractability of available reagents. Designed Ankyrin Repeat Proteins (DARPins) are a class of synthetic protein binder derived from diverse libraries by in vitro selection and tested by high-throughput screening to produce specific binders. In order to generate a functionally diverse toolset for studying inhibitory synapses, we screened a DARPin library against gephyrin mutants representing both phosphorylated and dephosphorylated states. We validated the robust use of anti-gephyrin DARPin clones for morphological identification of gephyrin clusters in rat neuron culture and mouse brain tissue, discovering previously overlooked clusters. This DARPin-based toolset includes clones with heterogenous gephyrin binding modes that allowed for identification of the most extensive gephyrin interactome to date and defined novel classes of putative interactors, creating a framework for understanding gephyrin's nonsynaptic functions. This study demonstrates anti-gephyrin DARPins as a versatile platform for studying inhibitory synapses in an unprecedented manner.

*For correspondence:
tyagarajan@pharma.uzh.ch

## Editor's evaluation

This article describes and validates new tools to study gephyrin biology in the brain, a critical regulator of synaptic inhibition and metabolism. The experiments are compelling, carefully controlled, and lead to a fundamental advance in neuroscience. This article will be of interest to a broad range of neuroscientists including those in synaptic, cellular, and circuit areas.

## Introduction

Biological research has relied for decades on the accuracy and precision of specific antibodies to morphologically describe protein localization and dynamics, or to biochemically describe protein interaction partners, using techniques such as immunolabeling, immunoprecipitation, and immuno-assays, among others. While antibody-based tools have been invaluable, for a given protein we often lack a variety of binders that perform excellently across applications. Antibodies that detect fixed proteins in tissue (which are typically partially denatured) may not bind with the same affinity or specificity to the same protein in a lysate (which may retain a more native confirmation). The heterogeneous quality of some commercial antibodies presents an additional challenge as the often ambiguous or

unknown antibody sequence, provenance, and specificity of poly- and monoclonal antibodies alike lead to false information and ultimately a high additional cost to research (*Bradbury and Plückthun, 2015*; *Smith, 2015*). This problem is especially relevant for the study of synaptic proteins, be they receptors or scaffolds, as these proteins are often used as markers to define the presence, plasticity, and regulation of synapses as a strong correlate for synaptic function. For example, ionotropic glutamate receptor subunits and the scaffolding molecule PSD-95 are frequently used to define the excitatory postsynapse, while GABA$_A$ receptors (GABA$_A$Rs) and the scaffolding protein gephyrin define the inhibitory postsynapse (*Micheva et al., 2010*).

Gephyrin is a highly conserved signaling scaffold that oligomerizes into multimers and binds to cognate inhibitory synaptic proteins to functionally tether GABA$_A$Rs at postsynaptic sites in apposition to presynaptic GABA release sites (*Tyagarajan and Fritschy, 2014*). Gephyrin is composed of three major domains: the N-terminal G domain and C-terminal E domain facilitate self-oligomerization of gephyrin underneath inhibitory postsynaptic sites, and they are linked together by the C domain, which is a substrate for diverse posttranslational modifications (*Sander et al., 2013*; *Tyagarajan and Fritschy, 2014*). Gephyrin mediates its scaffolding role by coordinating the retention of inhibitory synaptic molecules (*Figure 1A*), including GABA$_A$ and glycine receptors (GABA$_A$Rs, GlyRs), collybistin, and neuroligin 2 through interactions at locations within the E domain or E/C domain interface (*Choii and Ko, 2015*; *Tyagarajan and Fritschy, 2014*), with additional protein interactors binding to the G and C domains. These protein interactions can act synergistically to enhance postsynaptic density assembly and alter gephyrin lattice compaction (*Groeneweg et al., 2018*). Therefore, via homo- and heterophilic protein–protein interactions, gephyrin can control inhibitory postsynaptic function.

Gephyrin's scaffolding role is dynamically regulated by its post-translational modifications (PTMs). Gephyrin phosphorylation at several defined serine residues controls gephyrin oligomerization and compaction, thereby affecting GABAergic transmission (*Battaglia et al., 2018*; *Ghosh et al., 2016*; *Petrini and Barberis, 2014*; *Zacchi et al., 2014*). Two of these phosphosites, serines S268 and S270, are targeted by the kinases ERK1/2 and GSK3ß or cyclin-dependent kinases (CDKs), respectively, to downregulate gephyrin clustering (*Figure 1B*), thereby controlling postsynaptic strength (*Tyagarajan et al., 2013*). These phosphorylation events directly regulate gephyrin conformation via packing density changes to alter GABA$_A$ receptor dwell time (*Battaglia et al., 2018*), by altering gephyrin interacting partners (*Zhou et al., 2021*), or some combination of the two (*Specht, 2019*). Unfortunately, the most widely used anti-gephyrin antibody for identifying inhibitory postsynaptic sites, monoclonal antibody clone Ab7a, is sensitive to phosphorylation at serine 270 (*Kalbouneh et al., 2014*; *Kuhse et al., 2012*; *Zhou et al., 2021*), thus complicating interpretation of inhibitory postsynaptic presence, size, or dynamics.

In addition to PTMs, gephyrin is regulated by alternative splicing by a suite of exonic splice cassette insertions (annotation outlined in *Fritschy et al., 2008*). While the principal (P1) isoform of gephyrin in neurons facilitates its synaptic scaffolding role, gephyrin is also a metabolic enzyme that participates in molybdenum cofactor (MOCO) biosynthesis (*Nawrotzki et al., 2012*; *Schwarz and Mendel, 2006*; *Tyagarajan and Fritschy, 2014*). MOCO synthesis can be mediated in non-neuronal cells by an isoform that includes the C3 splice cassette (*Licatalosi et al., 2008*; *Meier et al., 2000*; *Smolinsky et al., 2008*), suggesting that gephyrin harbors both isoform- and cell-type-specific functions.

Gephyrin has been reported to complex with a wide variety of proteins as determined by both targeted and unbiased interaction studies (*Fuhrmann et al., 2002*; *Sabatini et al., 1999*; *Uezu et al., 2016*). These screens have implicated gephyrin in nonsynaptic processes, including regulation of mTOR signaling (*Sabatini et al., 1999*; *Wuchter et al., 2012*), and motor protein complexes (*Fuhrmann et al., 2002*). Furthermore, these interactomes have identified novel proteins such as InSyn1, with implications for understanding the heterogeneity of inhibitory synapse organization (*Uezu et al., 2019*). Still, the overlap in coverage of gephyrin's interactome in each study has been variable with respect to identification of canonical inhibitory synaptic proteins due to limitations of each screening technique. Taken together, there is a need to generate and characterize molecular tools that can (1) interrogate gephyrin in different applications, (2) be functionally validated for the experiment in question, and (3) be diverse enough in their mode of interaction to not limit the different protein functional states that can be probed.

Designed Ankyrin Repeat Proteins (DARPins) represent an attractive alternative tool compared to conventional antibodies as they are highly stable and specific synthetic protein binders that can be

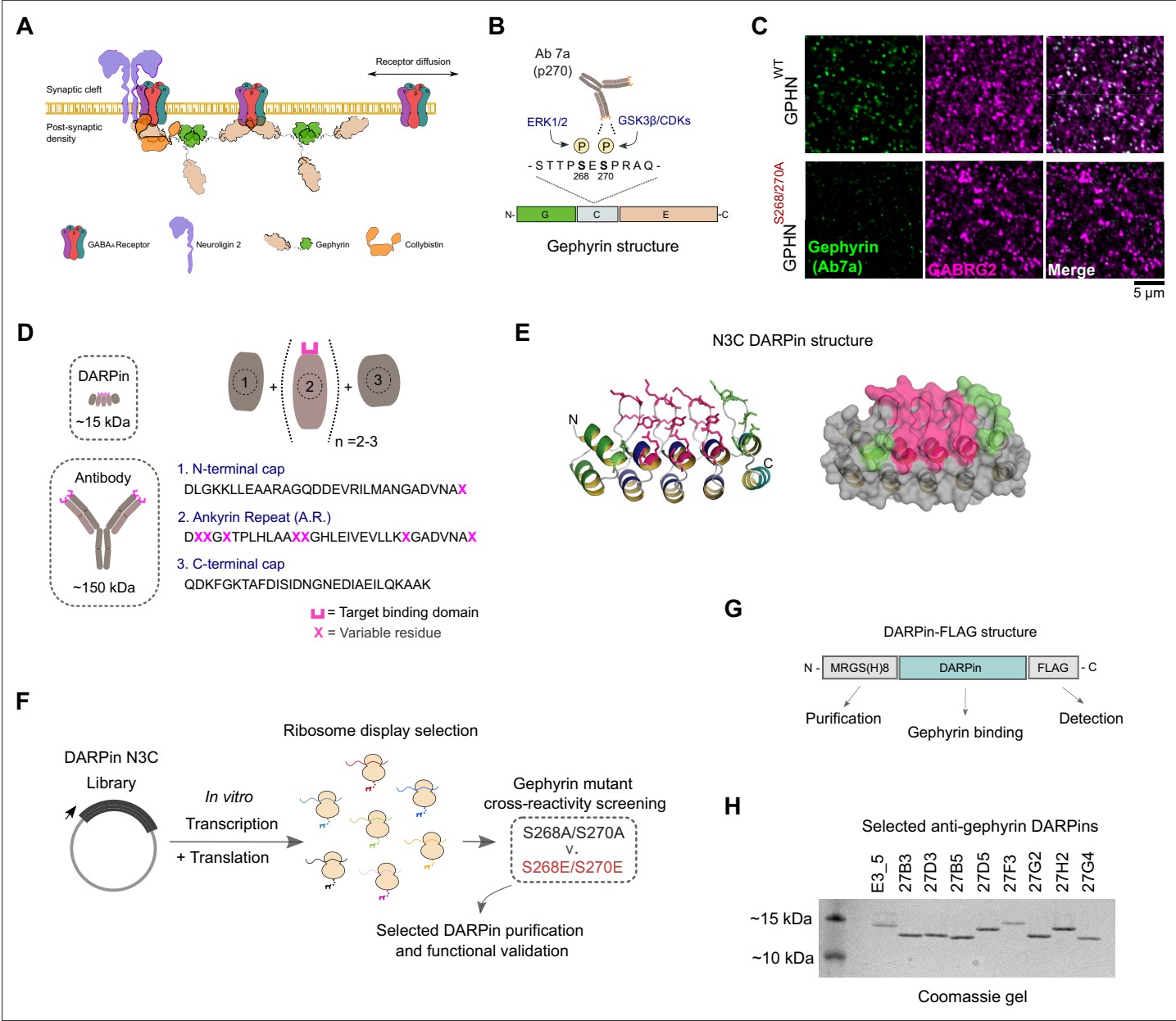

**Figure 1.** In vitro selection and generation of anti-gephyrin DARPins. (**A**) Diagram of gephyrin function at the inhibitory postsynapse via its scaffolding role. (**B**) Gephyrin domain structure and location of key phosphoserine residues S268 and S270, the commonly used antibody clone for detection of gephyrin (Ab7a) is phospho-S270-specific. (**C**) The antibody Ab7a does not detect gephyrin clusters colocalized with the γ2 GABA_A receptor subunit (GABRG2) in a phospho-null mouse model where S268 and S270 are mutated to alanines. (**D**) DARPins are an order of magnitude smaller than conventional antibodies and achieve target binding specificity by varying the sequence of ankyrin repeats (A.R.) with variable residues (magenta). (**E**) DARPin library design, with residues in magenta randomized in the original design and additional residues randomized in the caps (green). An N3C structure is shown with the N-terminal cap as a green ribbon and the C-terminal cap as a cyan ribbon with green side chains. (**F**) Schematic of anti-gephyrin DARPin selection and screening. (**G**) Structure of DARPin-FLAG clones used for initial validation experiments contain an N-terminal His_8 tag and C-terminal FLAG tag for purification and detection, respectively. (**H**) Coomassie-stained gel of the nonbinding control (E3_5) and eight anti-gephyrin DARPin binders.

The online version of this article includes the following source data and figure supplement(s) for figure 1:

**Source data 1.** Raw image and annotated uncropped Coomassie gel from *Figure 1H*.

**Figure supplement 1.** ELISA binding evaluation of anti-gephyrin DARPins.

**Figure supplement 2.** Sequence alignment of characterized anti-gephyrin DARPins.

generated via high-throughput in vitro selection and screening (*Binz et al., 2004*; *Kohl et al., 2003*). Since they possess a defined genetic sequence, they can be adapted into diverse fusion constructs, and their structural stability facilitates their engineering to achieve differential binding (*Harmansa and Affolter, 2018*; *Plückthun, 2015*). DARPins are composed of a variable number (typically 2–3) ankyrin repeats containing randomized residues, flanked by N- and C-terminal capping repeats with a hydrophilic surface that shield the hydrophobic core. Each repeat forms a structural unit, which consists of a β-turn followed by two antiparallel α-helices and a loop reaching the turn of the next repeat. The randomized residues on adjacent repeats within the β-turn turns and on the surface of the α-helices form a variable and contiguous concave surface that mediates specific interactions with target proteins. Using a DARPin library with high diversity (~$10^{12}$ unique DARPins), DARPins can be selected using ribosome display and then screened for particular binding characteristics (*Dreier and Plückthun, 2012*; *Douthwaite and Jackson, 2012*). Using this approach, DARPins have been shown to selectively bind to different conformations of proteins, including those brought about by phosphorylation (*Kummer et al., 2012*; *Plückthun, 2015*).

Despite being used extensively as both experimental tools for structural biology as well as therapeutics (*Plückthun, 2015*; *Tamaskovic et al., 2012*), DARPins have not yet been applied to neuroscience research in the current literature. In order to generate a new toolset of anti-gephyrin binders, we screened a DARPin library for binding to different gephyrin phosphorylation mutants and characterized the resulting DARPins in both morphological and biochemical applications. We validated the use of anti-gephyrin DARPins to understand how different binders can reveal novel aspects of gephyrin and inhibitory synapse biology highlighting heterogeneity of inhibitory postsynapse morphology and composition.

## Results

### Generation and selection of anti-gephyrin DARPins

Gephyrin clusters GABA$_A$ receptors and other inhibitory molecules such as neuroligin 2 and collybistin at postsynaptic sites (*Figure 1A*), where its clustering role is modified by phosphorylation, importantly at serines S268 and S270 (*Figure 1B*). This phosphorylation of gephyrin links upstream signaling (e.g., neurotrophic factors, activity) to downstream gephyrin regulation of inhibitory synaptic function (*Groeneweg et al., 2018*; *Tyagarajan and Fritschy, 2014*). The commonly used commercial antibody clone for morphological detection of synaptic gephyrin (clone Ab7a) has been employed extensively for almost four decades in the literature to identify inhibitory synapses (*Pfeiffer et al., 1984*). Though, rather than binding gephyrin regardless of its modified state, this antibody was recently demonstrated to specifically recognize gephyrin phosphorylated at serine S270 (*Kuhse et al., 2012*). This antibody's specificity for phospho-gephyrin complicates interpretation of synaptic gephyrin cluster identification when using clone Ab7a and prevents accurate detection of postsynaptic gephyrin clusters when gephyrin S270 phosphorylation is low or blocked. This is illustrated by the lack of binding of Ab7a to gephyrin in brain tissue derived from a phospho-S268A/S270A phospho-mutant mouse line, in which serines S268 and S270 are mutated to alanines (*Figure 1C*). Therefore, to generate protein binders that can more robustly identify gephyrin independently of its phosphorylation status, we looked beyond antibody-based binders to (DARPins).

DARPins are small (~12–15 kDa) compared to conventional antibodies (*Figure 1D*), and their binding to specific target proteins is mediated by several randomized residues contained within assemblies of 2–3 variable ankyrin repeats (AR) flanked by capping repeats (*Binz et al., 2004*). This basic DARPin structure creates a rigid concave shape with enhanced thermostability (*Figure 1E*). In addition, DARPins do not contain cysteines, allowing for functional cytoplasmic recombinant expression in *Escherichia coli* as well as cytoplasmic expression and functional studies in mammalian cells. We performed a ribosome display selection, followed by screening of individual clones against recombinant gephyrin (P1 principal isoform) containing either S268A/S270A or S268E/S270E mutations (*Figure 1F*), which mimic the respective dephosphorylated and phosphorylated state, thus representing functionally distinct gephyrin conformations (*Battaglia et al., 2018*; *Tyagarajan et al., 2013*). This allowed us to define sensitivity toward the modified state and widen the spectrum of DARPins obtained from the selection. Single DARPin clones were expressed in *E. coli* containing an N-terminal MRGS(H)$_8$ (His$_8$) tag and C-terminal FLAG tag (*Figure 1G*). Initial screening was performed with

376 DARPin clones using a high-throughput HTRF assay with crude extracts derived from 96-well expression plates. Of the initial hits, 32 were sequenced and 25 unique DARPins identified. These DARPins were further screened using an ELISA-based assay for relative binding to the phospho-null or phospho-mimetic gephyrin isoforms, or the absence of target as control (*Figure 1—figure supplement 1*). From this screen, eight DARPins were chosen for expression/purification and further analysis due to their high signal-to-background characteristics, as well as for equal binding to both phospho-mutant forms of gephyrin (*Figure 1H*, *Figure 1—figure supplement 1*). These eight DARPins showed diversity in the variable residues in the target protein interaction surface, highlighting the broad spectrum of binders that were obtained with this technology, and suggesting that they likely interact with gephyrin using different binding orientation or epitopes and independent of phosphorylation (*Figure 1—figure supplement 2*).

## Characterization of anti-gephyrin DARPins as morphological tools

The antibody clone Ab7a has been used extensively to both define the location, size, and dynamics of postsynaptic gephyrin puncta (*Bausen et al., 2010*; *Kalbouneh et al., 2014*; *Niwa et al., 2019*). However, this antibody reacts preferentially with gephyrin phosphorylated at S270, and sometimes also labels nonspecific structures such as the nucleus (*Figure 2A*). Alternative anti-gephyrin antibodies exist such as clone 3B11, which can be used for immunoprecipitation of gephyrin and detection on immunoblots, but leads to high background when used to label synapses (*Figure 2—figure supplement 1B*). To determine whether anti-gephyrin DARPins function as antibody-like tools in synaptic staining (in addition to binding recombinant gephyrin in vitro), we compared FLAG-tagged anti-gephyrin DARPins against antibody clone Ab7a for staining in primary rat hippocampal neuron culture at 15 days in vitro (DIV) (*Figure 2—figure supplement 1A–C*). While the unselected control DARPin clone E3_5-FLAG (*Binz et al., 2003*) did not present with detectable signal (*Figure 2A*), DARPin-FLAG clones 27B3, 27D3, 27F3, and 27G2 labeled gephyrin puncta with high specificity (*Figure 2A*, *Figure 2—figure supplement 1C*). Clone 27D5-FLAG produced no detectable signal, and clones 27B5, 27H2, and 27G4 labeled gephyrin puncta but produced considerable background comparable to another commercial anti-gephyrin antibody (clone 3B11) (*Figure 2—figure supplement 1A, B*). Moreover, clones 27B3, 27D3, 27F3, and 27G2 colocalized with presynaptic vesicular GABA transporter (VGAT)-containing axon terminals (*Figure 2B*). We compared the fraction of detected gephyrin puncta colocalized with VGAT, as well as the size of detected gephyrin clusters, using both the antibody Ab7a and selected DARPin-FLAG clones that displayed low background, namely, 27B3, 27D3, 27F3, and 27G2 (*Figure 2C and D*). We found no differences between DARPin-FLAG 27B3 or 27G2 and Ab7a colocalization with VGAT, indicating equal functionality in morphological applications. DARPin-FLAG 27D3 and 27F3 labeled puncta of a smaller size, which could relate either to their affinity for synaptic gephyrin or heterogeneity in epitope accessibility as different postsynaptic gephyrin puncta may differ in their isoform or post-translationally modified state.

## Anti-gephyrin DARPin-hFc fusion construct identifies phosphorylated and nonphosphorylated gephyrin clusters in mouse brain tissue

Identification of inhibitory synapses often involves the co-labeling of both pre- and postsynaptic structures using multiple antibodies raised in different species. In order to label gephyrin clusters in the brain, we replaced the His$_8$ and FLAG epitope tags from DARPin-FLAG clones 27B3, 27F3, 27G2, and the control clone E3_5 with an N-terminal human serum albumin (HSA) leader sequence and C-terminal human Fc (hFc) tag for mammalian recombinant production and purification and detection (*Figure 3—figure supplement 1*). The addition of the hFc tag allows for use in tandem with essentially all primary antibodies targeting synaptic markers raised in commonly used species such as rat, mouse, rabbit, goat, and guinea pig. Furthermore, it makes the construct bivalent. Consistently, DARPin-hFc 27G2 specifically labeled gephyrin puncta apposed to presynaptic VGAT terminals in both hippocampal neuron culture and mouse brain tissue (*Figure 3—figure supplement 2*). The specificity of this labeling could be confirmed by incubating DARPin-hFc 27G2 with a molar excess of recombinant gephyrin as a competitor, which led to the loss of immunofluorescent signal (*Figure 3—figure supplement 3*).

A significant fraction of synaptic gephyrin clusters are phosphorylated at serine 270, and therefore lead to an uncertain interpretation when their size and dynamics are assessed using the

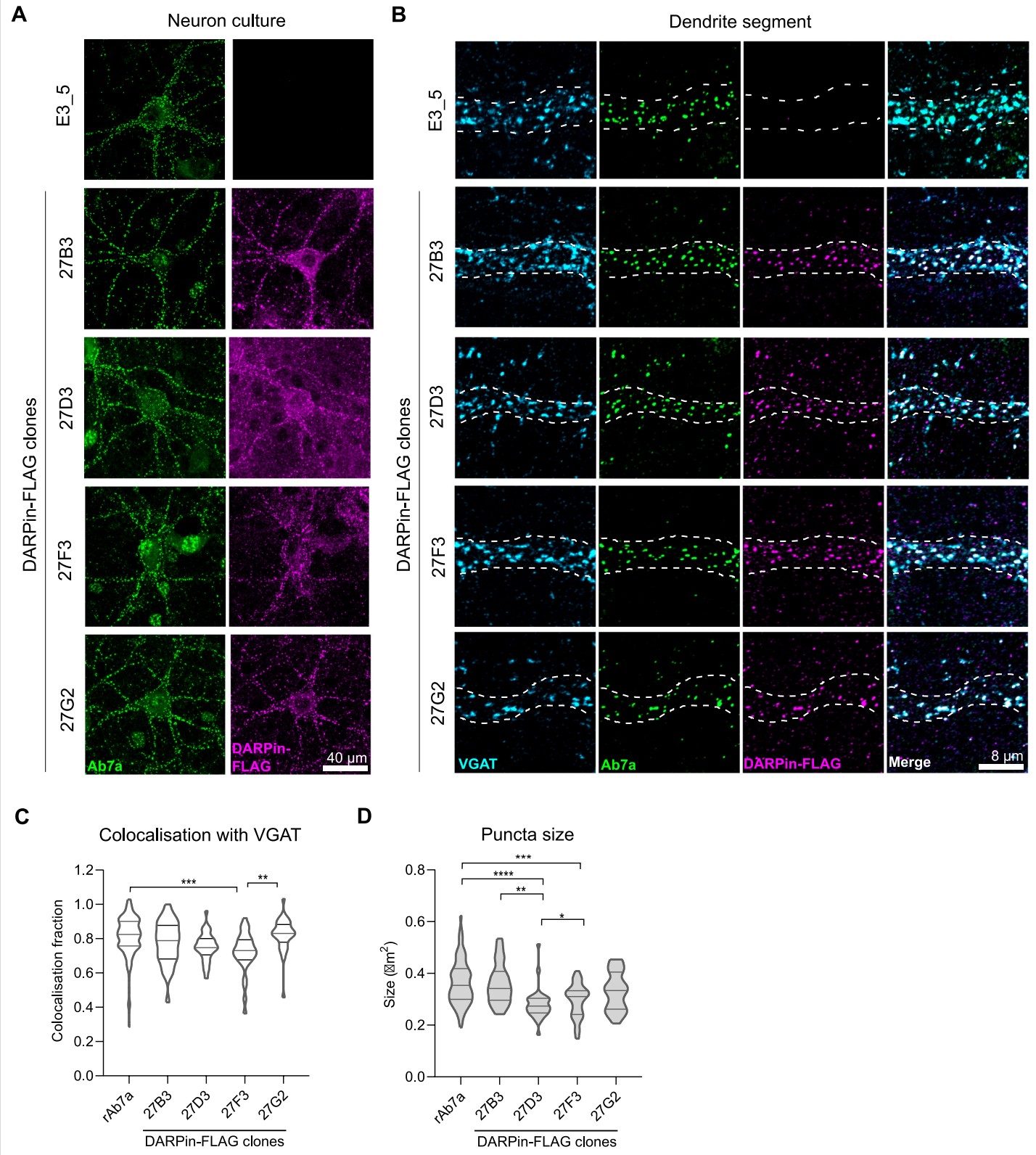

**Figure 2.** Anti-gephyrin DARPins specifically label gephyrin at inhibitory postsynaptic sites. Native gephyrin in fixed hippocampal neuron cultures (DIV15) probed using DARPin-FLAG clones, subsequently detected with anti-FLAG antibodies, and compared to staining with commercial anti-gephyrin antibody clone Ab7a. (**A**) Representative images of DARPin-FLAG clones 27B3, 27D3, 27F3, and 27G2 gephyrin puncta colocalized to Ab7a signal compared to the control DARPin E3_5. (**B**) Higher-magnification images of dendrite segments showing detected DARPin-FLAG signal colocalized with

*Figure 2 continued on next page*

*Figure 2 continued*

presynaptic VGAT. (**C**) Colocalization analysis indicating the fraction of gephyrin puncta that colocalize with VGAT along a proximal dendrite segment (>30 neurons/group pooled across three experiments). (**D**) Average puncta size identified by antibody Ab7a or DARPin-FLAG clones averaged by cell (pooled across neurons, >1100 synapses/group pooled across three experiment). Statistics**:** (**C, D**) one-way ANOVA, Tukey's post-hoc test comparing all groups ****$p<0.0001$, ***$p<0.0005$, **$p<0.005$, *$p<0.05$.

The online version of this article includes the following source data and figure supplement(s) for figure 2:

**Source data 1.** Data and statistical analysis to generate the violin plot in *Figure 2C and D*.

**Figure supplement 1.** Morphological characterization of DARPin-FLAG labeling in hippocampal neuron culture.

phospho-specific antibody Ab7a (*Kalbouneh et al., 2014*; *Specht, 2019*; *Zhou et al., 2021*). As predicted, DARPins-hFc 27G2 can label gephyrin puncta in both wildtype and phospho-S268A/S270A mutant mouse tissue while the commercial pS270-specific antibody Ab7a does not (*Figure 3A*).

The relative amount of Ab7a to anti-gephyrin DARPin signal could be used as a proxy to estimate relative gephyrin S270 phosphorylation at synapses. Indeed, we found that the Ab7a signal varied considerably both between adjacent synapses within a neuron and between neurons (*Figure 3B*, *Figure 3—figure supplement 4*). We confirmed the phosphosensitivity of this analysis method by inhibiting CDKs (upstream of gephyrin S270 phosphorylation) using 5 μM aminopurvalanol A applied for 24 hr. This treatment reduced Ab7a but not DARPin-hFc 27G2 signal as indicated by the decrease in the ratio between these two intensities seen both for individual synapses and when averaged by neuron (*Figure 3C and D*). We therefore examined the Ab7a/DARPin-hFc 27G2 intensity ratio between the somatic, dendritic, and axon-initial segment (A.I.S.) compartments in primary hippocampal neuron culture (*Figure 3E and F*), finding a significant reduction in Ab7a signal within the A.I.S. as defined by AnkyrinG immunolabeling (AnkG). Our results demonstrate that gephyrin phospho-S270 status varies between two neighboring clusters within a dendrite segment and also for the first time we can label gephyrin within the A.I.S. To test whether application of anti-gephyrin binders may affect the quantification of gephyrin clusters, we transfected hippocampal neuron cultures with EGFP-gephyrin plasmid and quantified cluster size along the principal dendrites of neurons using fluorescent signal from EGFP. This analysis demonstrated that compared to the untreated condition, application of either DARPin-hFc clones or antibody Ab7a does not influence the median size of EGFP gephyrin clusters in fixed tissue (*Figure 3—figure supplement 5*).

## DARPin-hFc 27G2 detects previously overlooked gephyrin clusters in brain tissue

Antibody-based identification of gephyrin clusters in the brain is widely used to identify inhibitory synaptic sites, but current reagents may only capture a subset of synaptic gephyrin clusters, namely, those with gephyrin significantly phosphorylated at S270. Therefore, we extended our analysis of postsynaptic gephyrin clusters using DARPin-hFc 27G2 and the phospho-S270-specific antibody Ab7a to mouse brain tissue using the hippocampal CA1 area as a model. The hippocampus is organized in a layered structure, stratifying somatic from dendritic compartments, with compartment-specific GABAergic interneuron innervation patterns well described (*Pelkey et al., 2017*). We found lamina-specific variability in relative gephyrin phosphorylation at S270, which was significantly elevated in the stratum oriens and stratum lacunosum moleculare compared to other layers (stratum pyramidale and radiatum) (*Figure 4A–C*). Within the stratum pyramidale, we noticed a population of large, relatively hypophosphorylated clusters (*Figure 4D*, *Figure 4—figure supplement 1*) reminiscent of A.I.S. synapses (*Figure 4E*). Indeed, while DARPin-hFc 27G2 labels large gephyrin clusters apposed to presynaptic VGAT terminals, Ab7a reactivity within the A.I.S. is relatively weak (*Figure 4F*). These hypophosphorylated clusters colocalize with the α2 GABA$_A$ receptor subunit thought to be enriched at the A.I.S. (*Lorenz-Guertin and Jacob, 2018*) and span the length of the A.I.S. as defined by the marker AnkG. Therefore, DARPin-hFc 27G2 can better assess postsynaptic gephyrin at the A.I.S. and at synapses where gephyrin phosphorylation is low. These data indicate that gephyrin clusters on the A.I.S. have likely gone un- or underreported in the literature, which is meaningful when considering that threshold-based detection of gephyrin is used as a proxy for inhibitory synapse presence and function (*Micheva et al., 2010*; *Schneider Gasser et al., 2006*).

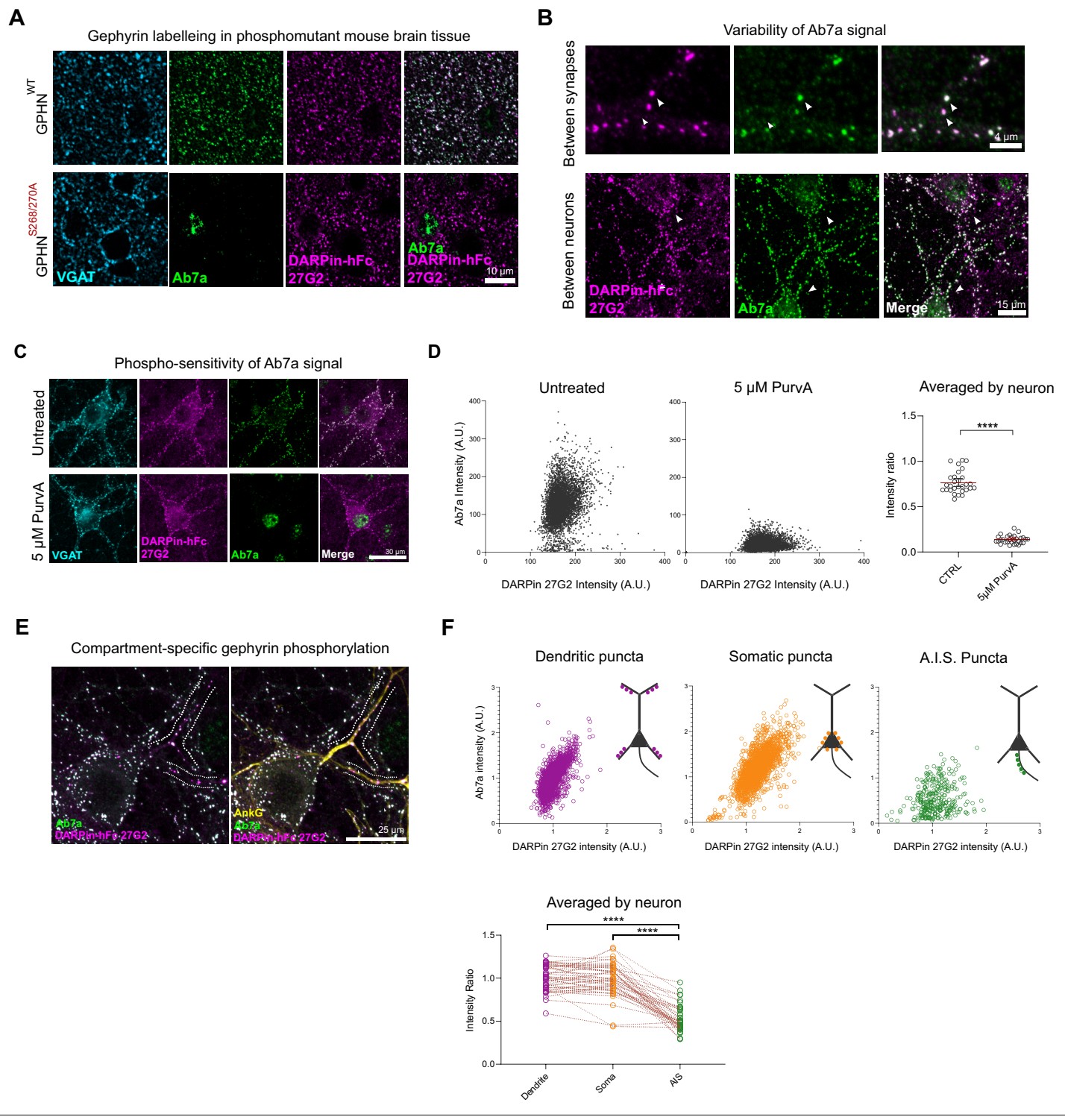

**Figure 3.** Phospho S270-insensitive DARPin-hFc 27G2 multiplexed with antibody Ab7a can assess synapse-specific gephyrin S270 phosphorylation.
(**A**) Representative images of DARPin-hFc 27G2 (but not antibody Ab7a) labeling gephyrin puncta in both wildtype (WT) and phospho-mutant gephyrin S268A/S270A mutant mouse brain tissue (somatosensory cortex layer 2/3). (**B**) Representative images from hippocampal neuron culture showing the relative Ab7a signal (indicating S270 phosphorylation) varies by synapse and between neurons. (**C**) Representative image showing DARPin-hFc 27G2 binding at synaptic puncta in primary hippocampal neuron culture is preserved after inhibition of CDKs following 24 hr treatment with 5 μM aminopurvalanol (PurvA) while Ab7a staining is severely reduced. (**D**) The relative fluorescence intensity at individual synapses (pooled from 30 neurons per group) showing a pronounced decrease in the average Ab7a/DARPin-hFc 27G2 intensity ratio. Quantification of Ab7a/DARPin-hFc 27G2

*Figure 3 continued on next page*

*Figure 3 continued*

fluorescence signal averaged across cells pooled from three independent experiments, n = 30 cells/group. (**E**) Representative images of hippocampal neuron culture used for quantification of relative Ab7a/DARPin-hFc labeling of clusters on the soma, proximal dendrites, or the axon-initial segment (A.I.S.) (AnkG). (**F**) Ab7a/DARPin intensity ratio of individual synapses pooled from 45 cells over three independent experiments showing a decrease in A.I.S. cluster Ab7a staining. Lower: quantification indicates significantly reduced A.I.S. Ab7a labeling of clusters compared to dendritic or somatic compartments. Statistics: (**D**) one-way ANOVA; (**F**) repeated-measures one-way ANOVA. All panels: *p<0.05, **p<0.01, ***p<0.001, ****p<0.0001. Mean and SD are presented.

The online version of this article includes the following source data and figure supplement(s) for figure 3:

**Source data 1.** Values and statistical results used to generate *Figure 3D and F*.

**Figure supplement 1.** Structure of DARPin-hFc 27G2.

**Figure supplement 2.** Validation of DARPin-hFc 27G2 for immunostaining.

**Figure supplement 3.** Competition with recombinant gephyrin reduces DARPin-hFc reactivity in tissue.

**Figure supplement 4.** Variation in Ab7a reactivity.

**Figure supplement 4—source data 1.** Values used to plot *Figure 3—figure supplement 4*.

**Figure supplement 5.** Anti-gephyrin binders do not alter EGFP gephyrin cluster size.

**Figure supplement 5—source data 1.** Values and statistical analysis used to plot *Figure 3—figure supplement 5*.

While gephyrin phosphorylation at S268 and S270 is thought to reduce gephyrin cluster size (*Tyagarajan et al., 2013*), the phosphosensitivity of clone Ab7a has prevented our analysis of this relationship as this antibody does not react with dephosphorylated gephyrin (S270 phosphorylation is blocked in the mutant mouse). Therefore, we applied DARPin-hFc 27G2 to analyze gephyrin clusters in both WT and our phospho-null S268A/S270A mutant mouse model (GPHN$^{S268A/S270A}$) (*Figure 4H and I*). We found that the median gephyrin cluster size is highest in the stratum oriens and stratum lacunosum moleculare in both WT and mutant mice, but that the median gephyrin cluster size is significantly enhanced across all layers when gephyrin phosphorylation is constitutively blocked in the S268A/S270A mutant mice (*Figure 4J*). This represents the first confirmation that native gephyrin clusters in the brain are importantly regulated by serine 268 and 270 phosphorylation. Moreover, the identification of layer- and compartment-specific gephyrin phosphorylation in the hippocampus indicates that the use of DARPin-hFc binders may be a more robust morphological tool to investigate the heterogeneity of gephyrin and inhibitory synapses in the brain.

## Multiple gephyrin protein complex precipitations using unique DARPin binders establish a consensus gephyrin interactome

Beyond applications for morphological detection of proteins in tissue, antibodies are essential for isolation of target protein complexes to understand their functional interaction networks. However, a network discovered by one binder may be different from another binder either due to affinity or epitope accessibility involving targets in specific functional states. Gephyrin was first identified as a scaffolding protein, and yet throughout the past decades has been implicated additionally in complex signaling processes mediated by changes in its ability to interact with different protein partners. To gain a more complete picture of gephyrin binding partners, we precipitated native gephyrin protein complexes from mouse brain lysates with the traditionally used antibody clone 3B11 (suitable for immunoprecipitations) and each one of our DARPin-hFc clones 27B3, 27F3, 27G2, and the control DARPin E3_5 (*Figure 5—figure supplement 1*). We then subjected the precipitated gephyrin complexes to interactor identification using quantitative liquid chromatography tandem mass spectrometry (LC-MS/MS) and compared the resulting interactomes (*Figure 5A and B*). We considered proteins to be present when they were detected using at least two peptide signatures. Furthermore, we considered proteins as part of gephyrin complexes when they were present either only in the binder condition, or at least a log$_2$ >2.5-fold enriched in the binder condition over the control DARPin E3_5 with a false discovery rate (FDR)-adjusted p-value cut-off under 0.05 (*Figure 5B*). These thresholds allow for a wider coverage to encompass most known interactors (*Figure 5—figure supplement 2*) such as collybistin (ARHG9), GABA$_A$ receptor subunits (GBRA1, 2), and a list of gephyrin interactors identified via BioID labeling (*Uezu et al., 2016*). Our results demonstrated that the abundance of canonical interactors spanned several orders of magnitude (*Figure 5—figure supplement 2*) and

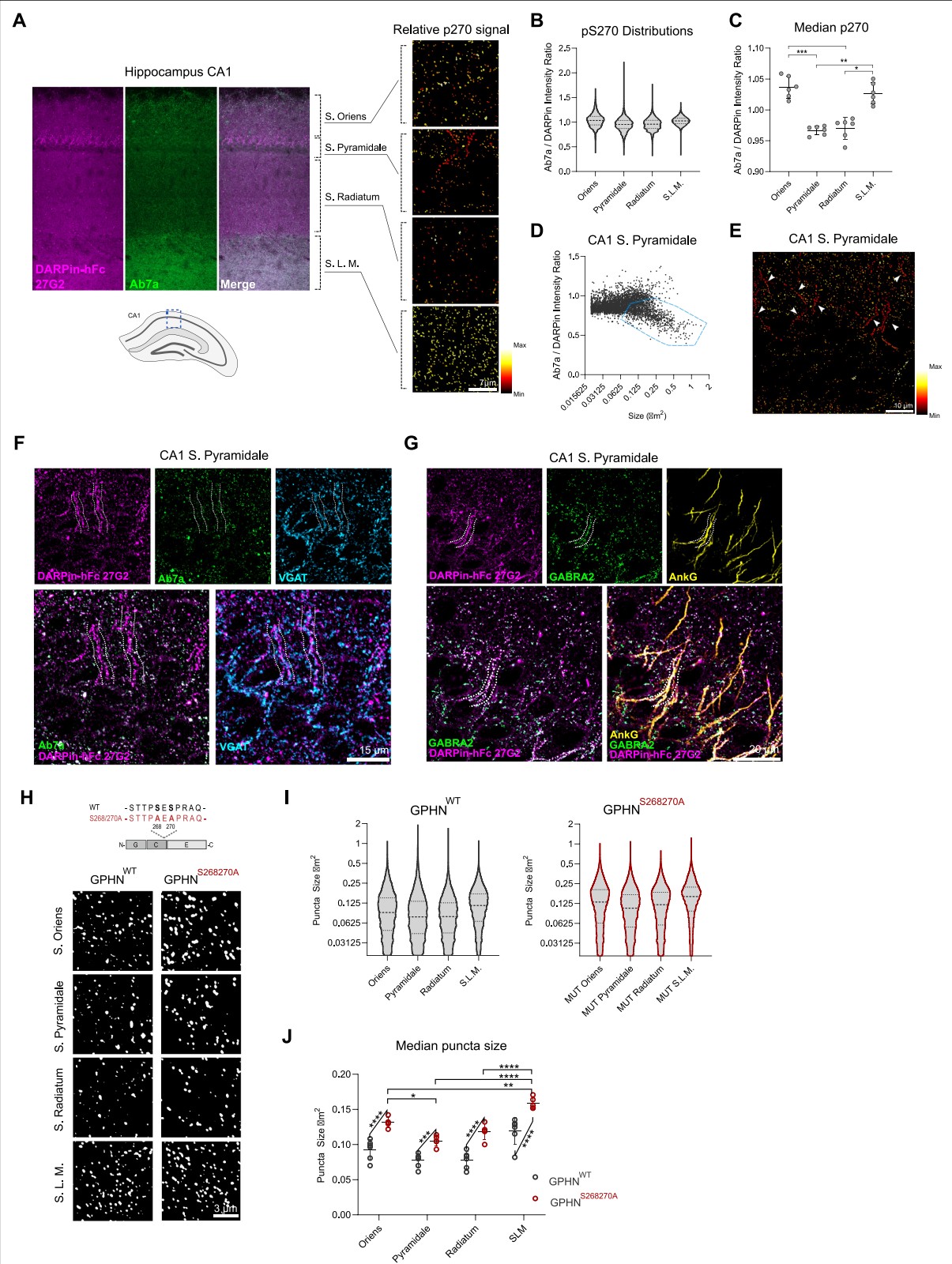

**Figure 4.** DARPin-hFc 27G2 labeling of gephyrin clusters demonstrates laminar and axon-initial segment (A.I.S.)-specific S270 phosphorylation and phosphorylation-dependent cluster size regulation. (**A**) Left: the relative Ab7a to DARPin-hFc 27G2 fluorescence intensity in the mouse hippocampus area CA1 shows layer-specific variability. Right: colorized gephyrin puncta indicating relative S270 phosphorylation as seen from hotter (more red/yellow) coloration. (**B**) Distribution of relative gephyrin phosphorylated at S270 (p270) at puncta between hippocampal lamina. Data pooled between

*Figure 4 continued on next page*

*Figure 4 continued*

six adult mice, three sections analyzed per mouse encompassing 14,000–47,000 gephyrin puncta per layer. (**C**) Analysis of the median relative gephyrin pS270 ratio between hippocampal layers (data pooled between sections per mouse, n = 6 mice quantified). (**D**) Example distribution of gephyrin pS270 signal by puncta size in the CA1 stratum pyramidale, with a population of large, hypophosphorylated clusters outlined. (**E**) Representative image of s. pyramidale with hot colors indicating gephyrin clusters with elevated phosphorylation; arrows indicate trains of large hypophosphorylated clusters. (**F**) Representative image showing large DARPin-identified gephyrin clusters apposed to presynaptic VGAT-containing terminals with corresponding low Ab7a antibody signal. (**G**) Representative image indicating gephyrin clusters on the A.I.S. (AnkG) colocalize with the $\alpha_2$ GABA$_A$ receptor subunit. (**H**) Representative images of gephyrin puncta identified using cluster analysis software in WT and S268A/S270A phospho-null mutant mice in the hippocampus using identical imaging parameters. (**I**) Violin plots indicating the distribution of gephyrin puncta sizes (14,000–47,000 puncta per group, pooled across 5–6 mice per group). (**J**) Analysis of the median puncta size between hippocampal layers and genotypes indicating larger gephyrin clusters in mutant mice. Statistics: (**C**) one-way ANOVA, (**J**) mixed-effects analysis comparing hippocampal lamina (horizontal bars) and genotypes (angled bars). All panels: $*p<0.05$, $**p<0.01$, $***p<0.001$, $****p<0.0001$. Median and SD are presented.

The online version of this article includes the following source data and figure supplement(s) for figure 4:

**Source data 1.** Data and statistical analysis presented in *Figure 4B–D, I, and J*.

**Figure supplement 1.** Relative pS270 synaptic distribution in the hippocampal CA1.

**Figure supplement 1—source data 1.** Data and statistical analysis presented in *Figure 4—figure supplement 1A, B*.

provided enhanced coverage compared to the previously established BioID-determined interactome (*Figure 5C*, *Figure 5—figure supplement 3*). Each interactome differed by the number of identified proteins (*Figure 5C*) where DARPin-hFc clones 27B3 and 27G2 identified 2–4 times more interactors than DARPin-hFc 27F3 or antibody 3B11, thus confirming the limitations of using only one binder to explore interacting protein networks.

High-confidence interactome determination is limited both by the sensitivity of interactor detection and the presence of false positives. Therefore, to compile a higher-confidence list of gephyrin interactors, we combined coverage between experiments using each DARPin-hFc clone to create a common gephyrin interaction network. We additionally cross-referenced this list with interactors precipitated by the antibody 3B11 as well as known binders identified from the literature to compile a high-confidence consensus gephyrin interactome (*Figure 5D*), representing the largest compilation of putative gephyrin interactors to date. This network encompasses the majority of canonical gephyrin-associated proteins, including GABA$_A$ and glycine receptors, inhibitory synaptic scaffolding and adhesion molecules, and cytoskeletal adaptor proteins. As expected, over-representation analysis of the consensus interactome found significant enrichment for synaptic organization processes, but also unexpectedly those involved in protein trafficking, mRNA regulation, and metabolic processes (*Figure 5—figure supplement 4*). Cataloging of individual proteins by functional ontology revealed clusters of gephyrin interactors in mRNA regulation, cytoskeletal proteins and adaptors, metabolic enzymes, and ribosomal subunits, together hinting at novel functions of gephyrin beyond synaptic scaffolding and MOCO biosynthesis (*Figure 5E*).

## Unique DARPin-hFc clones capture overlapping but ontologically distinct gephyrin interactomes

While our consensus gephyrin interactome may provide a robust framework to explore the related function of novel interacting proteins, the different scale of each network in terms of unique proteins identified and their different abundances suggests that each DARPin-hFc clone captures overlapping but unique gephyrin protein networks. To explore the extent of this phenomenon, we compared the relative abundance of interacting proteins that were constitutively present in all DARPin-hFc-derived gephyrin interactomes and identified a subset of proteins, which showed significant variation in the abundance between the three DARPin-hFc-based pulldowns (*Figure 6—figure supplement 1*). These included several canonical gephyrin interactors (*Figure 6A*). For example, clone 27F3 precipitated significantly more IQEC3 (a guanine nucleotide exchange factor important for synapse specification; *Früh et al., 2018*), while clone 27G2 captured gephyrin complexes containing more collybistin (ARHG9) (*Figure 6A*). Binder-specific protein abundance profiles were more pronounced when examining non-canonical gephyrin interactor sets such as metabolic enzymes, mRNA binding proteins, and ribosomal subunits. These ontology groups demonstrated a consistently higher abundance in clones 27B3 and 27G2 compared to 27F3-based gephyrin interactomes. This differential interactor

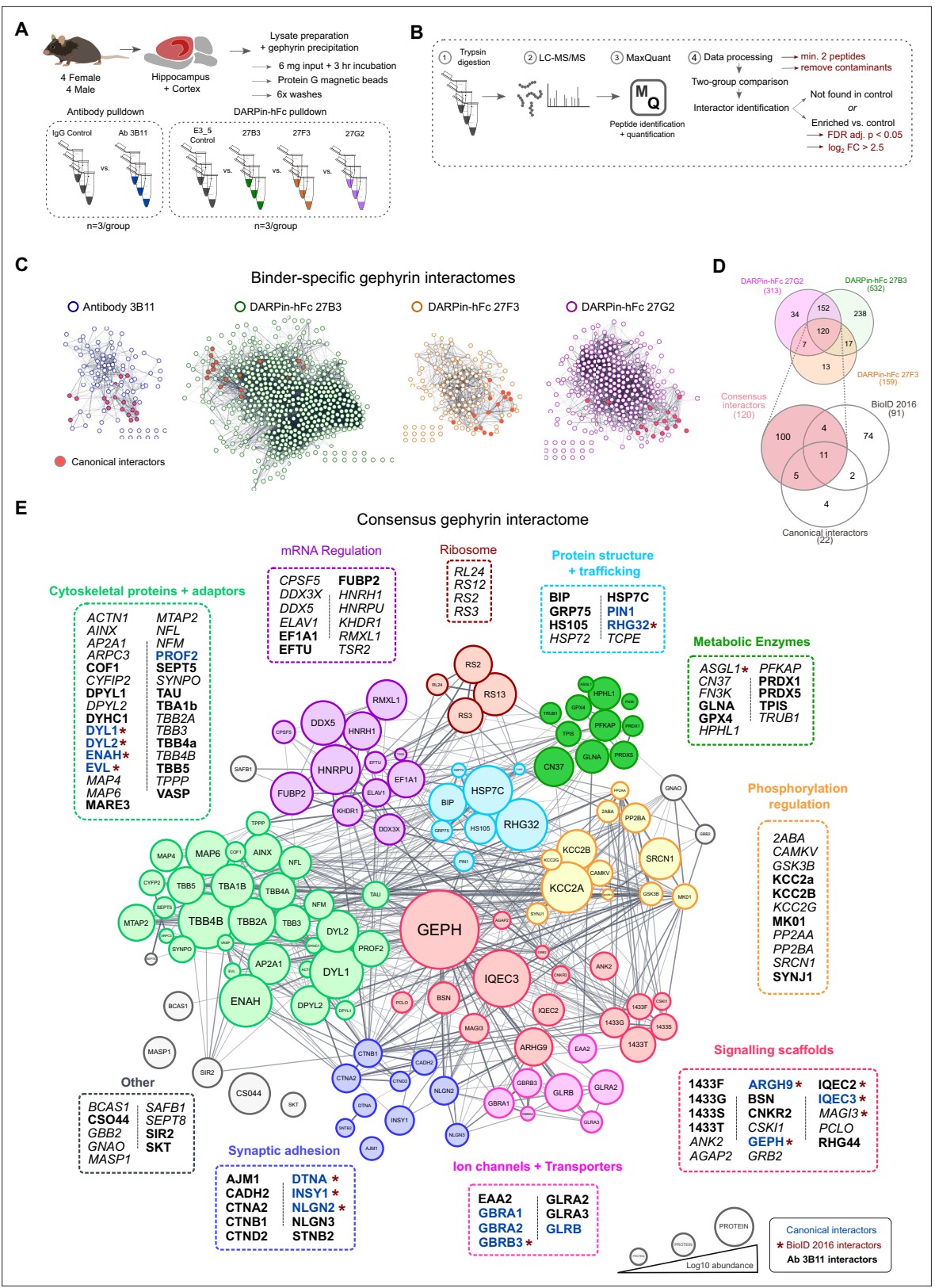

**Figure 5.** A DARPin-based consensus gephyrin interactome captures both known and novel protein interactors. **A)** Mouse brain tissue lysate preparation diagram. (**B**) Liquid chromatography tandem mass spectrometry (LC-MS/MS) and interactome determination methodology workflow indicating thresholds for consideration of interacting proteins. (**C**) Scale-free interaction networks (STRING) of gephyrin interactors identified from pulldowns using the commercial antibody 3B11, or DARPin-hFc 27B3, 27F3, and 27G2 compared to control conditions (containing antibody control

*Figure 5 continued*

IgG or the control DARPin-hFc E3_5). Nodes represent unique gephyrin interactors – red nodes indicate known (canonical) gephyrin interactors. (**D**) Venn diagram of the overlap in identified interactors from gephyrin complexes isolated using different DARPin-hFc clones; bottom indicates coverage compared to an extensive gephyrin interactome determined using BioID labeling (*Uezu et al., 2016*) and 22 canonical gephyrin interactors identified from the literature. (**E**) Consensus interactome of proteins identified by all DARPin-hFc clones and colored by protein ontology. Canonical gephyrin interacting proteins are indicated by blue font, and bold font indicates interactors also identified by the antibody clone 3B11. Asterisks indicate proteins previously identified by BioID (*Uezu et al., 2016*). Italic font indicates interactors exclusively identified by DARPins. Edges connecting protein nodes indicate putative interactions (STRING analysis), and node circle size indicates relative protein abundance averaged across all experiments.

The online version of this article includes the following source data and figure supplement(s) for figure 5:

**Source data 1.** List of interactors and relative abundance of detected proteins used to construct interaction networks and Venn diagrams in *Figure 5C–E*.

**Figure supplement 1.** Anti-gephyrin DARPins affinity purify gephyrin from mouse brain lysates.

**Figure supplement 1—source data 1.** Raw Coomassie gel images and immunoblots from *Figure 5—figure supplement 1*.

**Figure supplement 2.** Interactor identification plots.

**Figure supplement 2—source data 1.** Identity and quantification of abundance of interacting proteins presented in *Figure 5—figure supplement 2*.

**Figure supplement 2—source data 2.** Compiled list of proteins from all gephyrin interactor experiments used to assess gephyrin interactor identity.

**Figure supplement 3.** Interactome overlap with previous literature.

**Figure supplement 4.** Ontological enrichment analysis of the consensus gephyrin interactome.

abundance could be due either to DARPins interacting with functionally distinct isoforms of gephyrin or DARPin-specific interference with gephyrin conformation or interacting protein binding.

Gephyrin function is executed by several functional domains (G, C, and E domains), but it is also highly modified by phosphorylation as well as splice cassette insertions. To determine whether DARPin-hFc clones bind to different gephyrin domains or modified isoforms with different strength, we used an in-cell binding assay (*Figure 6—figure supplement 2*) to assess the relative binding of these clones to different forms of eGFP-tagged gephyrin. As expected from the in vitro characterization, there was no preference for any of the DARPin-hFc clones between wildtype gephyrin and the phospho-null or phospho-mimetic mutation-containing gephyrin at serines 268 and 270. Interestingly, we saw clear domain-specific binding preferences, with clones 27B3 and 27G2 interacting both with full-length gephyrin or the G and C domains in isolation, whereas clone 27F3 could only bind to full-length gephyrin (*Figure 6B*, *Figure 6—figure supplement 2*). Gephyrin splice cassette C3 is constitutively spliced out in neurons by the splicing factor NOVA (*Licatalosi et al., 2008*), implying it is not needed for synaptic scaffolding. However, the C3 cassette is included in gephyrin expressed within non-neuronal cells where it contributes toward MOCO synthesis activity (*Smolinsky et al., 2008*) or possible other functions (*Figure 6C*). We found that the C3 cassette is significantly less detected by DARPin-hFc 27F3, while clones 27B3 and 27G2 bind to both the principal (P1) and C3-containing cassette isoforms equally (*Figure 6C*, *Figure 6—figure supplement 2*). We additionally probed for binding to gephyrin containing the C4a cassette (thought to be brain-enriched but without a clearly identified function). None of the DARPin-hFc clones tested interacted strongly with the C4a-gephyrin isoform, while the antibody clone 3B11 interacted with this isoform at similar levels to the other gephyrin isoforms.

To understand whether the different DARPin-hFc clones can interact with ontologically distinct gephyrin protein networks, we performed over-representation analysis of proteins that are exclusive or significantly elevated in the interactome detected by clone 27F3 (neuronal isoform specific) or detected exclusively or significantly elevated by clones 27B3 and 27G2 (bind to neuronal and glial gephyrin isoforms). While we only saw enrichment for synaptic organization-related biological processes from DARPin-hFc 27F3-enriched interactors, we additionally found enrichment for cytoskeletal processes, ribosomal complex formation, and proteins involved in mRNA splicing and transport for the 27B3 and 27G2-enriched interactomes (*Figure 6D and E*). This suggests that the non-neuronal isoforms of gephyrin could be involved in these other distinct biological processes. In support of this hypothesis, when examining for proteins of glial or myelin ontology, we saw overall higher presence and abundance in the interactomes determined using clones 27B3 and 27G2 (*Figure 6—figure supplement 3*). We then confirmed DARPins' 27B3 and 27G2 bias for capturing gephyrin containing the C3 cassette by analyzing the relative abundance of C3-casette containing peptide fragments

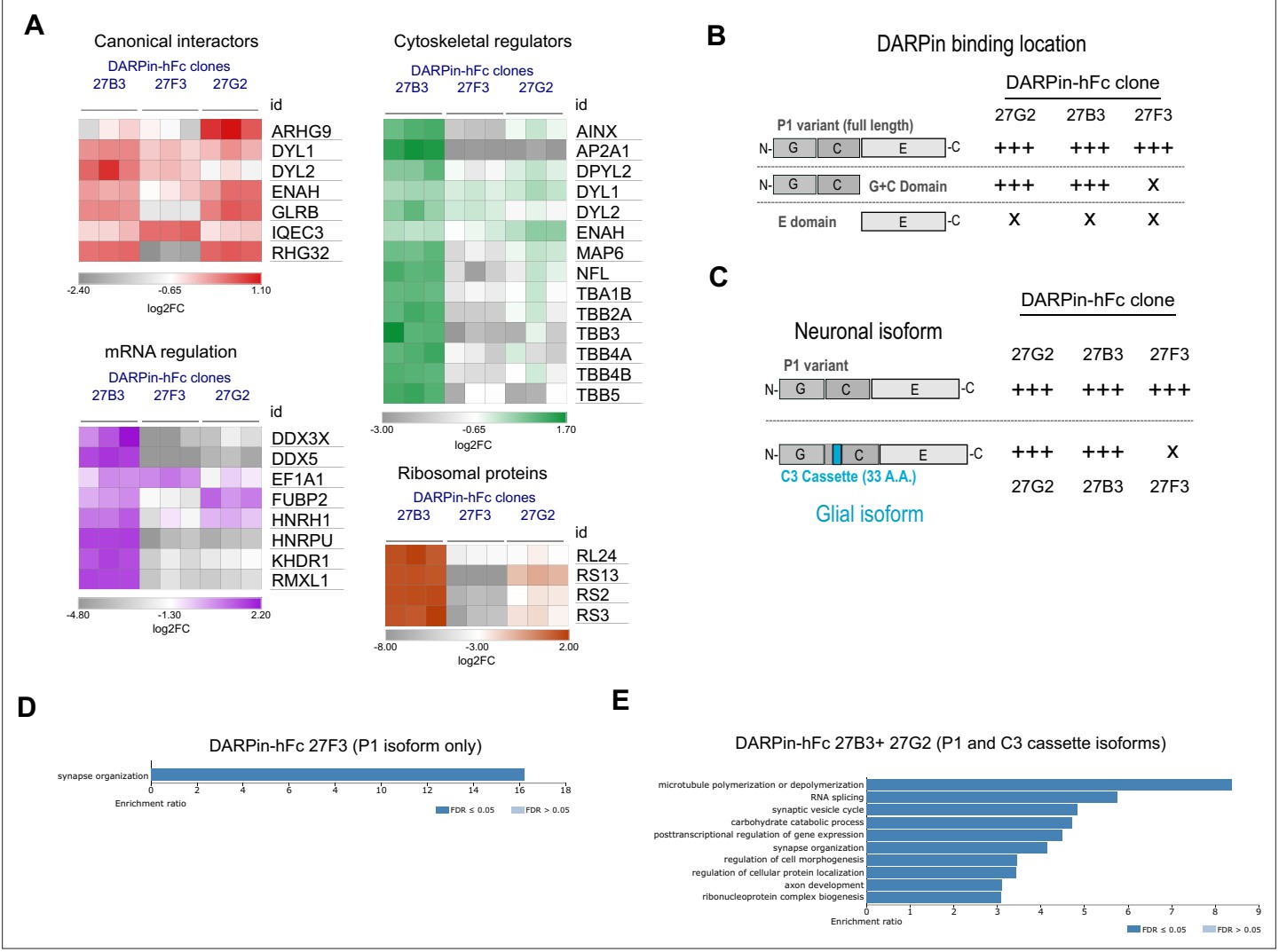

**Figure 6.** Diversity in DARPin-hFc clone-specific interactomes reveals putative isoform-specific gephyrin interactors. (**A**) Canonical and non-canonical (metabolic, mRNA binding, and ribosomal ontology) gephyrin interactors show binder-specific abundance profiles. Only significantly regulated interactors are shown. (**B**) DARPin-hFc clones 27B3 and 27G2 recognize both full-length gephyrin and the GC-domain while clone 27F3 recognizes only full-length gephyrin suggesting different binding epitopes. (**C**) DARPin-hFc 27F3 only recognizes the principal P1 (synaptic) isoform of gephyrin while clones 27B3 and G2 additionally recognize non-neuronal isoforms containing the C3 cassette. (**D**) DARPin-hFc 27F3-determined gephyrin interactome enriched over-representation analysis of biological processes. (**E**) DARPin-hFc 27B3 and 27G2-determined gephyrin interactome enriched over-representation analysis of biological processes. Statistics: (**A**) two-way ANOVA with multiple-comparisons correction comparisons all groups, three replicates per group.

The online version of this article includes the following source data and figure supplement(s) for figure 6:

**Source data 1.** Values used to generate heat maps in *Figure 6A*.

**Figure supplement 1.** DARPin-specific gephyrin interactor abundance.

**Figure supplement 1—source data 1.** Values and statistical test results indicating differentially abundant gephyrin interactors between binding experiments.

**Figure supplement 2.** Identification of gephyrin-binding preferences of anti-gephyrin DARPins using an in-cell HEK293T fluorescence assay.

**Figure supplement 2—source data 1.** Values and statistical analysis performed to generate graphs in *Figure 6—figure supplement 2B–D*.

**Figure supplement 3.** Non-neuronal interactor ontology.

**Figure supplement 3—source data 1.** Values used to generate heat maps in *Figure 6—figure supplement 3*.

**Figure supplement 4.** Relative C3 cassette recovery.

**Figure supplement 4—source data 1.** Values used to generate graphs in *Figure 6—figure supplement 4*.

recovered from pulldown experiments to total gephyrin, while clone 27F3 binds significantly less C3-casette containing gephyrin than other DARPins (*Figure 6—figure supplement 4*). These data indicate that understanding the isoform specificity of different DARPin clones will be useful for future dissection of gephyrin functionality at synapses, but also outside of synaptic sites or in non-neuronal cells.

## Discussion

In this study, we generated and characterized anti-gephyrin DARPins as tools to study inhibitory synapse biology. This novel class of gephyrin protein binder specifically interacts with gephyrin in both morphological and biochemical applications to label gephyrin clusters and isolate gephyrin protein complexes without the limitations of previous antibody-based tools. We furthermore demonstrated that these DARPins can capture a greater diversity of gephyrin forms, which will allow researchers to better characterize gephyrin and inhibitory synapses alike.

### Use of anti-gephyrin DARPins as morphological tools

Gephyrin is widely used as an inhibitory postsynaptic marker due to its specific enrichment at inhibitory postsynaptic sites, but current antibody epitope limitations mask the heterogeneity of postsynaptic gephyrin clusters that can be probed. As our DARPins are insensitive to modification at two key phospho-sites thought to be dynamically regulated at synapses, we were able to identify previously masked gephyrin clusters at the A.I.S. where relative gephyrin S270 phosphorylation is low (and thus difficult to detect with the antibody Ab7a). Because most image analysis methods use threshold-based detection of gephyrin cluster presence and dynamics, A.I.S. gephyrin clusters (and subsequently identification of inhibitory synapses) will be underrepresented in the literature. For example, by using only the antibody Ab7a, gephyrin was suggested to play a less important role in scaffolding A.I.S. synapses (*Gao and Heldt, 2016*), whereas the large gephyrin clusters illuminated using DARPins suggest the opposite. Inhibitory input onto the A.I.S. provided by Chandelier interneurons plays an important role in gating neuronal output (*Pelkey et al., 2017*). Therefore, studying gephyrin A.I.S. dynamics is especially relevant for uncovering mechanisms of network plasticity and how inhibition controls circuit function. Outside of the A.I.S., we documented clear changes in relative gephyrin S270 phosphorylation in the stratum oriens and stratum lacunosum moleculare of the CA1 region of the hippocampus, indicating potential interneuron-specific or input-layer-specific regulation of gephyrin function. Therefore, these DARPin-based tools can be used not only to robustly describe native gephyrin clusters in culture systems and in tissue, but they can also be used in tandem with gephyrin phospho-specific antibodies such as clone Ab7a to examine how genetics, environmental factors, or network activity regulate inhibitory adaptations via gephyrin. Moreover, DARPin binders may be able to better capture the heterogeneity of inhibitory postsynaptic sites that display differences in molecular composition regulation dependent on presynaptic inhibitory input (*Chiu et al., 2018*). The inclusion of the hFc tag on the DARPin constructs additionally allows them to be used with antihuman secondary antibodies, and thus in conjunction with the vast majority of commercial and homemade antibodies against other synaptic markers raised in nonhuman species.

DARPins lack cysteines, and thus have an advantage as protein binders over traditional antibodies as they can be expressed intracellularly as 'intrabodies' (*Plückthun, 2015*). Given their highly specific synaptic labeling, DARPin expression could be developed as a tool to visualize inhibitory synapses in living neurons or non-neuronal cells in vivo after by fusing DARPin clones to genetically encoded fluorescent proteins. The small genetic size of DARPins allows for their packaging along with additional elements such as inducible expression systems or other functional moieties into viral vectors with small genomic packaging limits. Future derivatization of anti-gephyrin DARPin binders, for example, using cell-type-specific drivers to express DARPins fused to different genetically encoded fluorescent proteins, could improve our understanding of how the inhibitory postsynapse remodels similarly or differentially within excitatory and inhibitory neurons within the same circuit after experimental intervention.

## Use of anti-gephyrin DARPins as biochemical tools

While gephyrin is used experimentally to morphologically identify the inhibitory postsynapse, it achieves its function through protein–protein interactions. Unbiased protein interaction network identification broadens how we envisage protein function and regulation. For example, a BioID-based gephyrin interactome discovered a novel inhibitory synaptic protein, InSyn1 (*Uezu et al., 2016*), which was found to be a key regulator of the dystroglycan complex and important for cognitive function (*Uezu et al., 2019*). By combining identified gephyrin interactors from antibody-based and DARPin-based experiments (including three distinct DARPin clones with different binding modalities), we were able to develop a consensus gephyrin interactome that facilitates higher-confidence pursuit of understanding how these proteins integrate or are regulated by gephyrin function. The thresholds and criteria used to identify gephyrin interactors were designed to be inclusive and are indeed able to capture a majority of established canonical gephyrin interactors, yet further assessment will be required to determine which interactors are functional, and additionally whether they interact with synaptic versus nonsynaptic gephyrin.

Various interactome determination techniques may capture different pictures of gephyrin protein networks. Proximity-ligation-based methods require expression of recombinant bait protein, which may not correspond to the endogenous expression level or diversity of isoforms of native proteins in cells, though they are able to capture transient interactions (*Burke et al., 2015*). Affinity purification of gephyrin protein complexes is more likely to capture stable gephyrin protein complexes and may not identify transient interactors, but it allows for identification of native gephyrin protein complexes reflecting the heterogeneity in its isoforms present or its post-translationally modified state. Therefore, using proximity-based labeling systems such as APEX and TurboID in conjunction with DARPins will allow for a comparison of stable (possibly structural) functions of gephyrin and transient (possibly signaling) roles of gephyrin.

A binder that interacts with gephyrin (whether antibody, DARPin, or other) may have the capacity to alter gephyrin interaction with protein partners. Future studies that directly determine the nature of binder–gephyrin interactions will allow us to better understand the extent of this phenomena. Due to the different binding modes of each DARPin to gephyrin, our consensus interactome offers a higher-confidence picture of which gephyrin interacting proteins are likely to occur regardless of how gephyrin itself is precipitated by a given binder.

Within our interactome data, we found previously unidentified but presumed interacting proteins that are well-known regulators of gephyrin. These include kinases such as CAMKIIα (KCC2A), which enhances gephyrin scaffolding via phosphorylation of serine 305 (*Flores et al., 2015*), GSK3β, which phosphorylates serine 270, and MK01 (ERK2), which targets serine 268 to reduce clustering (*Tyagarajan et al., 2013*), as well as Protein Phosphatase 2A, which antagonizes gephyrin phosphorylation at serine 270 (*Kalbouneh et al., 2014*). Additionally, we found the presence of multiple signaling scaffolds, including CNKR2, a PSD-associated protein that may regulate RAS-dependent MAPK signaling and is associated with intellectual disability in humans (*Hu et al., 2016*). This protein was very recently confirmed to regulate network excitability using a genetic model (*Erata et al., 2021*). These data suggest that many of the kinases known to regulate gephyrin scaffolding as well as regulators of those kinases are part of gephyrin protein complexes. Discovering how these kinase scaffolds associate and regulate gephyrin via phosphorylation may pave the way for targeted therapeutic development.

The name 'gephyrin' is derived from the Greek word γέφυρα meaning 'bridge' as it was discovered to link glycine receptors to the cytoskeleton (*Pfeiffer et al., 1982*; *Prior et al., 1992*), and subsequently found to interact with other cytoskeletal components, including dynein light chains 1 and 2 (*Fuhrmann et al., 2002*). We have now expanded this list to include multiple cytoskeletal interactors, including those involved in microtubule nucleation during cell division (e.g., TBG1, CENPV). Interestingly, gephyrin colocalized with microtubule nucleation centers has been recently identified in U2OS cells (*Zhou et al., 2021*).

Our consensus interactome identified not only canonical gephyrin binders but also unexpected proteins related to mRNA regulation, metabolism, and ribosomal function, which may suggest nonsynaptic functions of gephyrin yet to be described, the significance of which can now be investigated further with independent methods. Canonical gephyrin interactors differed in their abundance within complexes precipitated by clones that bind the P1 or C3 cassette variants, suggesting that different DARPin clones can access distinct synaptic gephyrin complexes. Gephyrin has been implicated

previously in regulation of mTOR, a signaling scaffold (*Machado et al., 2016*; *Sabatini et al., 1999*; *Wuchter et al., 2012*), as well as with elongation factor EF1A1, which, along with mTOR, directs mRNA translation and acts as a cytoskeletal adaptor complex (*Becker et al., 2013*). We identified EF1A1 as an interactor enriched in DARPin-precipitated complexes along with other mRNA binding proteins involved in mRNA splicing and transport (e.g., PURA, PURB, PABP1). Additionally, we detected the presence of transcription regulators such as SAFB1, DDX3X, and SIR2 from all DARPin complexes, and additional transcription factors, including MECP2 (a Rett syndrome-associated protein regulating inhibitory network development [*Pelkey et al., 2017*] and present at the PSD [*Aber et al., 2003*]) found only in 27B3 and 27G2 gephyrin complexes. Gephyrin signaling has recently been implicated in coupling transcriptional signaling via ARX in pancreatic beta cells (*Li et al., 2017*), and may therefore be involved in regulating additional transcriptional coupling in the brain via these described transcription factors. Many of the unexpected ribosomal and mRNA binding proteins were not detected in the control condition or using clone 27F3, suggesting that nonspecific binding to these classes of proteins is not an intrinsic property of DARPins. Further studies using isoform-specific DARPin clones to capture gephyrin protein networks in neuronal compared to non-neuronal cells will clarify which protein interactors may be isoform or cell-type-specific. Indeed, our group recently demonstrated that gephyrin affects microglial reactivity and synapse stability after stroke (*Cramer et al., 2022*).

## Further applications of DARPins

Beyond morphological and biochemical applications, DARPin binders can be developed further as functional tools. To date, no full-length experimentally determined gephyrin structural information exists, possibly due to the instability of gephyrin's C domain, making holo-gephyrin crystallization difficult (*Sander et al., 2013*), and developing approaches to stabilize gephyrin for structure determination is important to understand its structure–function relationship at the synapse (*Fritschy et al., 2008*). The stabilization of target proteins for structure determination has been a major experimental application of DARPins (*Batyuk et al., 2016*; *Tamaskovic et al., 2012*; *Wu et al., 2018*). In this study, we identified one DARPin clone (27F3) that binds only to the full-length P1 isoform but not individual domains. Using structural biology to assess the interaction between DARPins and full-length gephyrin, we may not only be able to rationally engineer DARPins to achieve different binding functionality, but may also derive fundamental information about gephyrin's form and function relationships, which would be essential for any future therapeutic efforts targeting gephyrin.

## Importance and limitations of protein binder development for neuroscience

Several synthetic protein binder scaffolds exist, including DARPins, nanobodies, anticalins, affibodies, and others (*Harmansa and Affolter, 2018*), providing a plethora of platforms to develop tools that detect or modify synaptic proteins, yet their application in neuroscience has lagged behind other fields. Of note, a fibronectin-based scaffold was used to generate intrabodies (termed FingRs by the authors) against gephyrin and the excitatory postsynaptic scaffold protein PSD-95 (*Gross et al., 2013*). This system has been used chiefly to label gephyrin clusters in living neurons (*Crosby et al., 2019*; *Gross et al., 2016*; *Son et al., 2016*; *Uezu et al., 2016*), but has not been extensively used for morphological detection of native gephyrin in tissue. Therefore, our DARPin-based toolset complements previously developed tools for live imaging, and future studies will test whether DARPins may be similarly used for native gephyrin tagging in living neurons. As with any protein binder that interacts with a substrate, DARPins have the potential to alter normal gephyrin interactions, clustering, or function. In the context of fixed tissue, we have demonstrated that this is not the case, though whether DARPins alter gephyrin clustering and inhibitory neurotransmission in living neurons requires evaluation before they can be used in this capacity.

Due to their stability and structure, DARPins are facile and inexpensive to produce and purify using simple bacterial systems and affinity resins. In addition, DARPins have relatively small sizes and defined sequences, which makes them experimentally tractable. We have shown that developing multiple DARPins to examine gephyrin is a useful strategy for understanding the heterogeneity of its signaling and function, and similar strategies applied to other synaptic beyond gephyrin are likely to yield fruitful insights, as previously demonstrated with other systems (*Plückthun, 2015*). For synaptic biology, these DARPins offer an additional toolset that we hope will be expanded in the future so that

excellent and well-characterized binders are available to probe a multitude of targets with the goal of enhancing research efficiency and facilitating discoveries.

# Materials and methods

## Key resources table

| Reagent type (species) or resource | Designation | Source or reference | Identifiers | Additional information |
|---|---|---|---|---|
| Recombinant DNA reagent | GST within 3′ 6xHis Tag | Provided by the UZH High Throughput Binder Selection platform | pET20b-A(H6)-GST | Used for subcloning recombinant gephyrin constructs for recombinant bacterial expression for use in the ribosome display selection |
| Recombinant DNA reagent | GST within 3′ 6xHis Tag and Avi tag | Provided by the UZH High Throughput Binder Selection platform | pET20b-A(H6)-AviTag | Used for subcloning recombinant gephyrin constructs for recombinant bacterial expression for use in the ribosome display selection |
| Recombinant DNA reagent | BirA enzyme | Provided by the UZH High Throughput Binder Selection platform | pBirAcm | Encodes the AVI-tag-specific biotin ligase BirA for biotin tagging of recombinant gephyrin constructs for use in the ribosome display selection |
| Recombinant DNA reagent | N-terminal 8xHis tag and C-terminal FLAG tag bacterial expression vector | Provided by the UZH High Throughput Binder Selection platform | pQIq_MRGS_HIS8_(DARPin)_FLAG | Used as the backbone for inserting DARPins using HindIII and BamHI restriction sites for recombinant bacterial expression of FLAG tagged DARPins |
| Recombinant DNA reagent | N-terminal HSA leader sequence and C-terminal hFc tag for mammalian expression | Provided by the UZH High Throughput Binder Selection platform | pcDNA3.1_SacB_hFc | Used as the backbone for inserting DARPins using HindIII and BamHI restriction sites for recombinant mammalian expression of hFc tagged DARPins |
| Recombinant DNA reagent | N-terminal His-tagged P1-gephyrin S268/270A | This article | pET20b-A(H6)- P1-gephyrin S268/270A | Subcloned from pEGFPC2-gephyrin S268/270A (*Tyagarajan et al., 2013*) using added Kpn1 and EcoRI sites into pET20b-A(H6)-GST for use in DARPin ribosome display selection |
| Recombinant DNA reagent | N-terminal His-tagged P1-gephyrin S268/270E | This article | pET20b-A(H6)- P1-gephyrin S268/270E | Subcloned from pEGFPC2-gephyrin S268/270E (*Tyagarajan et al., 2013*) using added Kpn1 and EcoRI sites into pET20b-A(H6)-GST for use in DARPin ribosome display selection |
| Recombinant DNA reagent | N-terminal His-tagged P1-gephyrin | This article | pET20b-A(H6)- P1-gephyrin | Subcloned from pEGFPC2-gephyrin P1 (*Tyagarajan et al., 2013*) using added Kpn1 and EcoRI sites into pET20b-A(H6)-GST for use in DARPin ribosome display selection |
| Recombinant DNA reagent | N-terminal HisAvi-tagged P1-gephyrin S268/270A | This article | pET20b-A(H6)- P1-gephyrin S268/270A AviTag | Subcloned from pEGFPC2-gephyrin S268/270A (*Tyagarajan et al., 2013*) using added Kpn1 and EcoRI sites into pET20b-A(H6)-AviTag for use in DARPin ribosome display selection |
| Recombinant DNA reagent | N-terminal HisAvi-tagged P1-gephyrin S268/270E | This article | pET20b-A(H6)- P1-gephyrin S268/270E AviTag | Subcloned from pEGFPC2-gephyrin S268/270E (*Tyagarajan et al., 2013*) using added Kpn1 and EcoRI sites into pET20b-A(H6)-AviTag for use in DARPin ribosome display selection |
| Recombinant DNA reagent | N-terminal eGFP-tagged P1-gephyrin S268/270A | *Tyagarajan et al., 2013* | pEGFPC2-gephyrin S268/270A | Used for subcloning for recombinant bacterial expression as well as the in-cell fluorescence assays |
| Recombinant DNA reagent | N-terminal eGFP-tagged P1-gephyrin S268/270E | *Tyagarajan et al., 2013* | pEGFPC2-gephyrin S268/270E | Used for subcloning for recombinant bacterial expression as well as the in-cell fluorescence assays |

*Continued on next page*

*Continued*

| Reagent type (species) or resource | Designation | Source or reference | Identifiers | Additional information |
|---|---|---|---|---|
| Recombinant DNA reagent | N-terminal eGFP-tagged P1-gephyrin | *Tyagarajan et al., 2013* | pEGFPC2-gephyrin P1 | Used for subcloning for recombinant bacterial expression as well as the in-cell fluorescence assays |
| Recombinant DNA reagent | N-terminal eGFP-tagged gephyrin GC domain | *Lardi-Studler et al., 2007* | EGFPC2-Gephyrin GC | Used for in-cell fluorescence assays to assess relative binding of DARPins to the GC domain of gephyrin |
| Recombinant DNA reagent | N-terminal eGFP-tagged gephyrin E domain | *Lardi-Studler et al., 2007* | EGFPC2-Gephyrin E | Used for in-cell fluorescence assays to assess relative binding of DARPins to the E domain of gephyrin |
| Recombinant DNA reagent | N-terminal eGFP-tagged gephyrin containing the C3 cassette | *Smolinsky et al., 2008* | pEGFPC2 Gephyrin C3 | Used for in-cell fluorescence assays to assess relative binding of DARPins to the C3 cassette containing gephyrin variants |
| Recombinant DNA reagent | N-terminal eGFP-tagged gephyrin containing the C4a cassette | *Smolinsky et al., 2008* | pEGFPC2 Gephyrin C4a | Used for in-cell fluorescence assays to assess relative binding of DARPins to the C4a cassette containing gephyrin variants |
| Recombinant DNA reagent | DARPin-FLAG E3_5 (control) | This article | pQIq_MRGS_HIS8_(E3_5)_FLAG | Created by subcloning DARPin E3_5 into pQIq_MRGS_HIS8_(DARPin)_FLAG using BamHI and HindIII sites |
| Recombinant DNA reagent | DARPin-FLAG 27B3 | This article | pQIq_MRGS_HIS8_(27B3)_FLAG | Created by subcloning DARPin 27B3 into pQIq_MRGS_HIS8_(DARPin)_FLAG using BamHI and HindIII sites |
| Recombinant DNA reagent | DARPin-FLAG 27D3 | This article | pQIq_MRGS_HIS8_(27D3)_FLAG | Created by subcloning DARPin 27D3 into pQIq_MRGS_HIS8_(DARPin)_FLAG using BamHI and HindIII sites |
| Recombinant DNA reagent | DARPin-FLAG 27F3 | This article | pQIq_MRGS_HIS8_(27F3)_FLAG | Created by subcloning DARPin 27F3 into pQIq_MRGS_HIS8_(DARPin)_FLAG using BamHI and HindIII sites |
| Recombinant DNA reagent | DARPin-FLAG 27B5 | This article | pQIq_MRGS_HIS8_(27B5)_FLAG | Created by subcloning DARPin 27B5 into pQIq_MRGS_HIS8_(DARPin)_FLAG using BamHI and HindIII sites |
| Recombinant DNA reagent | DARPin-FLAG 27D5 | This article | pQIq_MRGS_HIS8_(27D5)_FLAG | Created by subcloning DARPin 27D5 into pQIq_MRGS_HIS8_(DARPin)_FLAG using BamHI and HindIII sites |
| Recombinant DNA reagent | DARPin-FLAG 27G2 | This article | pQIq_MRGS_HIS8_(27G2)_FLAG | Created by subcloning DARPin 27G2 into pQIq_MRGS_HIS8_(DARPin)_FLAG using BamHI and HindIII sites |
| Recombinant DNA reagent | DARPin-FLAG 27H2 | This article | pQIq_MRGS_HIS8_(27H2)_FLAG | Created by subcloning DARPin 27H2 into pQIq_MRGS_HIS8_(DARPin)_FLAG using BamHI and HindIII sites |
| Recombinant DNA reagent | DARPin-FLAG 27G4 | This article | pQIq_MRGS_HIS8_(27G4)_FLAG | Created by subcloning DARPin 27G4 into pQIq_MRGS_HIS8_(DARPin)_FLAG using BamHI and HindIII sites |
| Recombinant DNA reagent | DARPin-hFc E3_5 (control) | This article | pcDNA3.1_ E3_5 _hFc | Created by subcloning DARPin E3_5 into pcDNA3.1_ SacB _hFc using BamHI and HindIII sites |
| Recombinant DNA reagent | DARPin-hFc 27B3 | This article | pcDNA3.1_27B3_hFc | Created by subcloning DARPin 27B3 into pcDNA3.1_ SacB _hFc using BamHI and HindIII sites |
| Recombinant DNA reagent | DARPin-hFc 27F3 | This article | pcDNA3.1_27F3_hFc | Created by subcloning DARPin 27F3 into pcDNA3.1_ SacB _hFc using BamHI and HindIII sites |

*Continued on next page*

*Continued*

| Reagent type (species) or resource | Designation | Source or reference | Identifiers | Additional information |
|---|---|---|---|---|
| Recombinant DNA reagent | DARPin-hFc 27G2 | This article | pcDNA3.1_27G2_hFc | Created by subcloning DARPin 27G2 into pcDNA3.1_ SacB _hFc using BamHI and HindIII sites |
| Recombinant DNA reagent | His/His-AVI F | Microsynth | 5'-A TAT GGT ACC CAC CAC CAC CAC CAC CAC TGA G-3' | Forward primer used to amplify gephyrin and gephyrin S268A/S270A or E mutants for insertion into recombinant expression vectors (His and His-AVI plasmids) |
| Recombinant DNA reagent | His-AVI R | Microsynth | 5'-T ATA GAA TTC TGA AGA GCC TCC TGA AGA GCC TCC TTC ATG CCA TTC-3' | Reverse primer used to amplify gephyrin and gephyrin S268/270A or E mutants for insertion into recombinant expression vectors (HIS-AVI plasmids) |
| Recombinant DNA reagent | His-R | Microsynth | 5'-T ATA GAA TTC TGA AGA GCC TCC TGA AGA GCC TCC GTG ATG GTG ATG GT-3' | Reverse primer used to amplify gephyrin and gephyrin S268A/S270A or E mutants for insertion into recombinant expression vectors (His- plasmids) |
| Antibody | Anti-Ankyrin G (AnkG) (mouse monoclonal) | Neuromab | MABN466; RRID AB_274980 | IF/ICC used at 1:1000 |
| Antibody | Anti-mouse AP (goat polyclonal) | Sigma-Aldrich (Merck) | A3562; RRID:AB_258091 | Used for ELISA screen |
| Antibody | Anti-FLAG M2 (mouse monoclonal) | Sigma-Aldrich (Merck) | F3165; RRID AB_259529 | IF/ICC used at 1:1000 |
| Antibody | Anti-FLAG D2 (mouse monoclonal) | Cisbio | 61FG2DLB | Used for HTRF screen |
| Antibody | Anti-GABRA2 (guinea pig polyclonal) | In-house (*Fritschy and Mohler, 1995*) | - | IF/ICC used at 1:2000 |
| Antibody | Anti-gephyrin 3B11 (mouse monoclonal) | Synaptic Systems | Cat# 147111; RRID:AB_887719 | IF/ICC used at 1:1000 |
| Antibody | Anti-gephyrin Ab7a (rabbit monoclonal) | Synaptic Systems | 147 008; RRID:AB_2619834 | IF/ICC used at 1:2000 |
| Antibody | Anti-VGAT (guinea pig monoclonal) | Synaptic Systems | 131308; RRID:AB_2832243 | IF/ICC used at 1:2000 |
| Antibody | Anti-mouse Alexa Cy3 (goat polyclonal) | Jackson ImmunoResearch Labs | JAC 115-165-166; RRID:AB_2338692 | IF/ICC used at 1:500 |
| Antibody | Anti-rabbit Alexa 488 (goat polyclonal) | Jackson ImmunoResearch Labs | JAC 111-545-144; RRID:AB_2338052 | IF/ICC used at 1:500 |
| Antibody | Anti-guinea pig Alexa 647 (goat polyclonal) | Jackson ImmunoResearch Labs | JAC 106-605-003; RRID:AB_2337446 | IF/ICC used at 1:500 |
| Antibody | Anti-human Cy3 (goat polyclonal) | Jackson ImmunoResearch Labs | JAC 109-165-170; RRID:AB_2810895 | IF/ICC used at 1:500 |
| Peptide, recombinant protein | Streptavidin-Tb cryptate | Cisbio | 610SATLB | Used for HTRF screen |
| Antibody | IRDye 680RD anti-mouse IgG (donkey polyclonal) | LI-COR Biosciences | LIC925-68072 | WB 1:20,000 |
| Antibody | Anti-human polyclonal Fc HRP | CalBiochem | 401455 | WB 1:40,000 |

*Continued on next page*

*Continued*

| Reagent type (species) or resource | Designation | Source or reference | Identifiers | Additional information |
|---|---|---|---|---|
| Cell line (*Rattus norvegicus*) | Wistar (RccHan:WIST) hippocampal cell culture | Envigo (Netherlands) | Order code: 168 | E17 embryos were collected from time-mated dams |
| Biological sample (*Mus musculus*) | Tissue C57BL/6JCrl | Charles River Laboratories (Germany) | RRID:IMSR_JAX:000664 | Used for synapse analysis and proteomic analysis |
| Biological sample (*M. musculus*) | Tissue C57Bl6/JCrl GphnS268A/S270A | *Cramer et al., 2022* | NA | Used for synapse analysis only |
| Strain, strain background (*Escherichia coli*) | BL21 DE3 Gold | Bio-Rad | Cat# 161-0156 | Used for recombinant bacterial gephyrin and DARPin expression |
| Strain, strain background (*E. coli*) | XL1-blue | Agilent | 200249 | Used for DARPin ribosome display screening |
| Cell line (human) | HEK293T | ATCC | CRL 11268 | Used for in-cell DARPin binding screen |

## Cloning and expression of gephyrin phosphorylation mutants

The principal (P1) rat isoform of gephyrin (referred to as wildtype [WT]) or the P1 variant containing mutated serine to alanine (phospho-null) or serine to glutamic acid (phospho-mimetic) mutations at serines 268 and 270 has been described previously (*Tyagarajan et al., 2013*). Primers introducing a 5′ EcoRI restriction site upstream of a 2× GSSS linker sequence and 3′ KpnI site (*see* Key Resources Table) were used to amplify WT or mutated gephyrin before restriction digest and ligation into target vectors for recombinant bacterial expression and purification containing a 5′ $His_8$ tag or His-Avi tag. *E. coli* BL21-DE3 Gold was transformed with the correct clones, and clones containing the His-Avi tag were transformed along with a plasmid encoding BirA for AviTag-specific biotin ligation. Bacteria were grown in THY media (20 g tryptone, 10 g yeast extract, 11 g HEPES, 5 g NaCl, 1 g $MgSO_4$/L pH 7.4) containing ampicillin (100 µg/mL) and chloramphenicol (10 µg/mL) to ensure expression of both tagged Avi-gephyrin and BirA. Overnight 5 mL cultures were used to inoculate a 150 mL culture grown at 37°C and 250 rpm until an $OD_{600}$ of 0.7 was reached. Induction and biotinylation were achieved by using a final concentration of 30 µM IPTG and 50 µM D-biotin (dissolved in 10 mM bicine buffer, pH 8.3). Protein induction proceeded for 6 hr before bacteria were pelleted.

Bacterial pellets were resuspended in 15 mL lysis buffer (50 mM Trizma base, 120 mM NaCl, 0.5% NP-40) containing cOmplete Mini Protease Inhibitor Cocktail (Roche) and DNAseI (Roche) before sonication on ice to release proteins. The lysate was pelleted at 20,000 × *g* at 4°C for 15 min, and the cleared lysate was passed through 0.45 and 0.22 µm sterile filters. $His_8$-tagged proteins were affinity purified on a 1 mL nickel agarose column (HIS-Select) using gravity flow. The lysate volume was passed 2× through the column then washed 1× with 6-column volumes of medium salt equilibration buffer (300 mM NaCl, 50 mM $NaH_2PO_4$, 10 mM imidazole, pH 8.0), then 1× with low-salt buffer (same with 100 mM NaCl), 1× with medium-salt buffer (300 mM NaCl), 1× with high-salt buffer (same with 500 mM NaCl), then 2× with medium-salt buffer (300 mM NaCl). Proteins were eluted in 4 mL elution buffer (equilibration buffer containing 250 mM imidazole) and dialyzed in storage buffer (150 mM NaCl, 50 mM $NaH_2PO_4$, pH 7.5) using dialysis tubing. Dialyzed protein was centrifuged at 60,000 × *g* to remove any aggregated products, and the concentration was determined using absorption at 280 nm using a NanoDrop spectrophotometer with predicted protein molecular weight and extinction coefficient values determined using ProtParam online software (ProtParam, Swissprot, https://web.expasy.org/protparam/). Protein biotinylation was assessed using a streptavidin shift assay and stored at –80°C.

## Anti-gephyrin DARPin selection and screening

To generate DARPin binders, biotinylated gephyrin S268E/S270E was immobilized alternately on either MyOne T1 streptavidin-coated beads (Pierce) or Sera-Mag neutravidin-coated beads (GE),

depending on the particular selection round. Ribosome display selections were performed essentially as described (*Dreier and Plückthun, 2012*) using a semi-automatic KingFisher Flex MTP 96-well platform. The library includes N3C-DARPins with stabilized C-terminal caps (*Kramer et al., 2010*). This library is a mixture of DARPins with randomized and nonrandomized N- and C- terminal caps, respectively (*Plückthun, 2015*; *Schilling et al., 2014*), and successively enriched pools were cloned as intermediates in a ribosome display specific vector (*Schilling et al., 2014*). Selections were performed over four rounds with decreasing target concentration and increasing washing steps to enrich for binders with high affinities. The first round included the initial selection against gephyrin S268E/S270E at low stringency. The second round included pre-panning with the opposite phospho-null (gephyrin S268A/S270A) variant immobilized on magnetic beads, with the supernatant transferred to immobilized target of the same variant. The third round included this pre-panning of the opposite variant and the addition of the (non-biotinylated) same variant to enrich for binders with slow off-rate kinetics. The fourth and final round included only the pre-panning step and selection was performed with low stringency.

The final enriched pool was cloned as fusion construct into a bacterial pQE30 derivative vector with an N-terminal MRGS(H)$_8$ tag (His$_8$) and C-terminal FLAG tag via unique BamHI × HindIII sites containing *lacIq* for expression control. After transformation of *E. coli* XL1-blue, 380 single DARPin clones for each target protein were expressed in 96-well format and lysed by addition of a concentrated Tris-HCL-based HT-Lysis buffer containing octylthioglucoside (OTG), lysozyme, and nuclease or B-Per Direct detergent plus lysozyme and nuclease (Pierce). These bacterial crude extracts of single DARPin clones were subsequently used in a homogeneous time-resolved fluorescence (HTRF)-based screen to identify potential binders. Binding of the FLAG-tagged DARPins to streptavidin-immobilized biotinylated gephyrin variants was measured using FRET (donor: streptavidin-Tb; acceptor: anti-FLAG-d2, Cisbio). Further HTRF measurement against 'No Target' allowed for discrimination of gephyrin-specific hits.

From the identified binders, 32 were sequenced and 25 unique clones were identified. The DARPins were expressed in small scale, lysed with Cell-Lytic B (Sigma), and purified using a 96-well IMAC column (HisPur Cobalt plates, Thermo Scientific). DARPins after IMAC purification were analyzed at a concentration of 10 µM on a Superdex 75 5/150 GL column (GE Healthcare) using an Aekta Micro System (GE Healthcare) with phosphate-buffered saline (PBS) containing 400 nM NaCl as the running buffer to identify monomeric DARPin binders. Final hit validation of specificity was performed by ELISA using small-scale IMAC-purified DARPins. Binding of the FLAG-tagged DARPins to streptavidin-immobilized biotinylated gephyrin variants was measured using a mouse-anti-FLAG-M2 antibody (Sigma) as first and goat-anti-mouse-alkaline phosphatase-conjugated antibody (Sigma) as second antibody. Further ELISA measurement against 'No Target' allowed for discrimination of gephyrin-specific hits. The best binders did not discriminate between phospho-mimetic states, suggesting that other epitopes were favored. A list of DARPin sequences for clones characterized in this study is available as *Supplementary file 1*.

## Cloning and recombinant expression of anti-gephyrin DARPins

Bacterial expression and purification of FLAG-tagged DARPins was performed as for His-tagged gephyrin constructs. Purification was validated using SDS-PAGE and Coomassie staining of acrylamide gels. Sub-cloning of select DARPins into a vector containing an N-terminal HSA leader sequence and C-terminal human Fc fragment (hFc) region using BamHI and HindIII restriction sites was performed for mammalian cell production. Test rounds of DARPin-hFc fusion expression were performed in adherent HEK293T cells where the supernatant was collected to confirm DARPin hFc expression. Medium-scale production of DARPin-hFc fusion constructs was performed with assistance from the Protein Production and Structure core facility (PTPSP Lausanne) by transfecting plasmids for clones 27B3-hFc, 27F3-hFc, and 27G2-hFc, as well as control DARPin E3_5-hFc into nonadherent HEK cells and grown in 400 mL cultures. DARPin-hFc recombinant protein was affinity-purified using Protein A resin after overnight incubation with rotation at 4°C and captured on a 15 mL column Protein A Sepharose resin (GenScript), then beads were washed with 50-column volumes of PBS and eluted with glycine buffer pH 3.0 into 1.5 M Tris–HCl pH 8.0 before overnight dialysis into PBS pH 7.5. Concentration was determined using a NanoDrop spectrophotometer using the A280 extinction coefficient.

## Gephyrin binding fluorescence assay in HEK293T cells

An in-cell fluorescence-based assay was developed to characterize the relative binding of anti-gephyrin DARPin clones to eGFP-tagged gephyrin variants in order to assess binding and validate the DARPin screening ELISA results in cells. HEK293T cells were acquired directly from ATCC and were validated by STR profiling and tested negative for mycoplasma contamination. HEK293T cells were maintained in DMEM with 10% FCS at 37°C in a 5% $CO_2$ jacketed incubator. Cells were seeded onto glass coverslips and grown to 50% confluency before transfecting plasmids (using standard PEI-based transfection at a ratio of 1 µg plasmid to 4 µg PEI). eGFP-tagged gephyrin P1 variant, as well as those containing serine-to-alanine or -glutamate mutations at S268 and S270 (S268A/S270A, S268E/S270E), has been previously described (*Tyagarajan et al., 2013*). eGFP-tagged gephyrin E domain or GC domains (*Lardi-Studler et al., 2007*) as well as variants containing the C3 or C4a splice cassettes *Lardi-Studler et al., 2007* have been described previously. Cells grown on coverslips were washed briefly in PBS and fixed in 4% paraformaldehyde (PFA) for 15 min. Coverslips were washed in PBS, then treated with 1:2000 (1 mg/mL stock) dilution of DARPin-FLAG clones or a control clone (nonbinding DARPin E3_5-FLAG) in 10% normal goat serum (NGS) for 90 min. Coverslips were washed and then treated with a 1:1000 dilution of mouse anti-FLAG antibody (clone M2, Sigma) for 60 min, then washed 3× in PBS. Coverslips were incubated with an Alexa 647-conjugated goat anti-mouse secondary antibody and DAPI for 30 min prior to washing 3× with PBS and drying before mounting with DAKO mounting medium onto glass slides.

Coverslips were imaged using an LSM700 microscope (Zeiss) with ×40 (1.4 NA) objectives. Images were acquired using Zen software (Zeiss). Laser intensity and gain settings were set to maximize signals in all channels/conditions without bleed-through or signal saturation, and acquisition settings were kept consistent for comparative analyses. eGFP-gephyrin-positive HEK cells were imaged at random locations on the coverslip, and fluorescent signals were acquired at 8 bits in the 488 and 647 channels to capture the eGFP-gephyrin and FLAG signal, respectively. eGFP-gephyrin presents as a diffuse signal in the soma with occasional cytoplasmic aggregates. For intensity analysis, regions of interest (ROIs) were manually drawn within the cytosol to avoid inclusion of these aggregates in the quantification. Fluorescence intensity was quantified using ImageJ. The slope of the relationship between the eGFP-gephyrin signal and the FLAG signal was used to compare relative binding of DARPins to their target.

## Animals

C56Bl/6J mice were purchased from Charles River (Germany), and timed-pregnant Wistar rats (for E17 embryo collection for neuron culture) were purchased from Envigo (Netherlands). The S268A/S270A phospho-null mouse was previously generated using CRISPR-Cas9 editing to mutate residues at the endogenous locus (*Cramer et al., 2022*). The collection of embryonic and adult tissue was performed in accordance with the European Community Council Directives of November 24, 1986 (86/609/EEC). Tissue collection was performed under license ZH011/19 approved by the Cantonal Veterinary Office of Zurich.

## Synaptic staining, imaging, and analysis

Hippocampal cell cultures derived from E17 Wistar rat embryos were prepared as previously described (*Tyagarajan et al., 2013*) containing a mixture of excitatory/inhibitory neurons and glia grown on poly-L-lysine-coated glass coverslips. Cultures were maintained for 15 DIV before use to allow for synapse formation. Neurons were prepared for DARPin-FLAG or DARPin-hFc staining and immunostaining as with HEK293T cultures, with the exception that endogenous gephyrin was analyzed using the anti-gephyrin antibody clone Ab7a (Sysy 147 011) or clone 3B11 (Sysy 147 111). Guinea pig anti-VGAT antibody (Sysy 131 004) and mouse anti-Ankyrin G (Neuromab, MABN466) were used to identify inhibitory presynapses and the A.I.S., respectively. Homemade affinity-purified guinea pig anti-GABRA2 was used to detect postsynaptic sites in tissue. Optimal concentrations of anti-gephyrin DARPins for staining were determined for each clone; 1:2000 dilution from 1 mg/mL stock was determined to be best for DARPin-FLAG, and 1:4000 dilution performed best for DARPin-hFc. For the analysis presented in *Figure 3—figure supplement 5*, neuron cultures were prepared as above, but transfected with EGFP-gephyrin plasmid at DIV 8 using Lipofectamine 2000 reagent as previously (*Tyagarajan et al., 2013*).

For brain tissue staining, animals were anesthetized with intraperitoneal injections of pentobarbital before trans-cardial perfusion with oxygenated, ice-cold artificial cerebrospinal fluid (ACSF: 125 mM NaCl, 2.5 mM KCl, 1.25 mM $NaH_2PO_4$, 26 mM $NaHCO_3$, 25 mM D-glucose, 2.5 mM $CaCl_2$, and 2 mM $MgCl_2$). Perfused brains were dissected and post-fixed in 150 mM PBS containing 4% PFA (pH 7.4) for 90 min at 4°C. Tissue was cryoprotected overnight in PBS containing 30% sucrose 4°C, then cut into 40-μm-thick sections using a sliding microtome. Sections were stored at –20°C in antifreeze solution (50 mM sodium phosphate buffer with 15% glucose, 30% ethylene glycol at pH 7.4) until use. For immunofluorescence experiments, sections were washed 3 × 10 min under gentle agitation in TBST (50 mM Tris, 150 mM NaCl, 1% Tween, pH 7.5) before overnight incubation in primary antibody solution (with or without DARPin inclusion) (TBST containing 0.2% Triton X-100 and 2% NGS). For DARPin-hFc 27G2, a concentration of 1:4000 was used (from 1 mg/mL stock). Sections were then washed 3 × 10 min and incubated for 30 min at room temperature with secondary antibodies in TBST solution with 2% NGS (Jackson). Sections were washed again 3 × 10 min in TBST before transfer to PBS and mounting onto gelatine-coated slides, then covered using DAKO mounting medium. For all tissue morphological analysis, image acquisition, processing, and analysis were acquired/performed blind to condition using identical imaging parameters. Images used for synapse quantification experiments were acquired on a Zeiss LSM 800 laser scanning confocal microscope operating Zen image acquisition software (Zen 2011) using ×63 oil immersion objectives (N.A. 1.4). Identical imaging settings were used when comparing between groups in a given experiment. Relative Ab7a/DARPin-hFc 27G2 fluorescent intensity cluster analysis was performed using the Analyse Particles functionality of Fiji after thresholding. Synaptic colocalization analysis was performed using a custom ImageJ macro previously described (*Panzanelli et al., 2017*).

## Precipitation of gephyrin complexes for LC-MS/MS interactome determination

Tissue lysates were prepared from acutely isolated cortexes and hippocampi of four male and four female C57BL/6J mice (Charles River) on ice and immediately homogenized in cold EBC lysis buffer (50 mM Tris–HCl, 120 mM NaCl, 0.5% NP-40, and 5 mM EDTA with cOmplete Mini Protease Inhibitors (Roche) and phosphatase inhibitor cocktails 2 and 3 (Sigma)) and incubated on ice for 60 min. Lysates were cleared by centrifugation at 20,000 × *g* for 20 min, and the supernatant protein concentration measured using a BCA assay. Gephyrin complexes were captured by incubating protein lysate (total 6 mg of protein per reaction) with DARPin-hFc binders or the control DARPin clone E3_5 or, control IgG, or 3B11 mouse-anti-gephyrin antibody for 3 hr at 4°C with rotation. In order to precipitate similar amounts of gephyrin protein, 4 μg of 3B11 antibody, or approximately 2 μg of anti-gephyrin DARPin-hFc (adjusted for equimolar concentration), was used per reaction (1.5 mL volume total). Complexes were precipitated using 20 μg of Protein G magnetic beads (30 min incubation with rotation) and washed 6× in 600 μL of EBC buffer. The supernatant was removed and replaced with 25 μL of PBS and immediately submitted for LC-MS/MS sample preparation.

## Immunoblotting

For immunoblotting experiments, input and precipitated samples were prepared in 5× SDS buffer containing beta-mercaptoethanol (Bio-Rad) and boiled for 5 min at 90°C. Protein concentration determination was performed using a BCA assay (Pierce). Acrylamide gels were either stained with Coomassie dye or transferred to PVDF membranes. Gephyrin was detected using a mouse anti-gephyrin antibody (clone 3B11, 1:1000), and DARPin-hFc was detected using an anti-hFc (HRP conjugated, 1:40,000) antibody overnight and detected using anti-mouse IR 680 dye (LI-COR) on a LI-COR imager or an HRP detection kit using a Fuji imager.

## On-bead digestion

Captured immunocomplexes were processed immediately after precipitation. Beads were washed once in 100 μL digestion buffer (10 mM Tris + 2 mM $CaCl_2$, pH 8.2). After resuspension in 45 μL digestion buffer, proteins were reduced and alkylated with 2 mM TCEP and 20 mM chloroacetamide, respectively, for 30 min at 60°C in the dark. 5 μL of Sequencing Grade Trypsin (100 ng/μL in 10 mM HCl, Promega) was added to the beads, and the digestion was carried out in a microwave instrument (Discover System, CEM) for 30 min at 5 W and 60°C. The supernatants were transferred into new

tubes and the beads were washed with 150 µL 0.1% TFA, then pooled with the previous supernatant. The samples were dried and resolubilized with 20 µL of 3% acetonitrile, 0.1% formic acid for MS analysis. Prior to MS analysis, the peptides were diluted to an absorption (A280) of 0.2.

## Liquid chromatography-mass spectrometry analysis

Mass spectrometry analysis was performed on an Orbitrap Fusion Lumos (Thermo Scientific) equipped with a Digital PicoView source (New Objective) and coupled to an M-Class UPLC (Waters). Solvent composition at the two channels was 0.1% formic acid for channel A and 0.1% formic acid, 99.9% acetonitrile for channel B. For each sample, 1 µL of diluted peptides was loaded on a commercial MZ Symmetry C18 Trap Column (100 Å, 5 µm, 180 µm × 20 mm, Waters) followed by nanoEase MZ C18 HSS T3 Column (100 Å, 1.8 µm, 75 µm × 250 mm, Waters). The peptides were eluted at a flow rate of 300 nL/min using a gradient from 5 to 22% B in 80 min, 32% B in 10 min, and 95% B for 10 min. The mass spectrometer was operated in data-dependent mode (DDA) acquiring a full-scan MS spectra (300–1500 *m/z*) at a resolution of 120,000 at 200 *m/z* after accumulation to a target value of 500,000. Data-dependent MS/MS spectra were recorded in the linear ion trap using quadrupole isolation with a window of 0.8 Da and HCD fragmentation with 35% fragmentation energy. The ion trap was operated in rapid scan mode with a target value of 10,000 and a maximum injection time of 50 ms. Only precursors with intensity above 5000 were selected for MS/MS, and the maximum cycle time was set to 3 s. Charge state screening was enabled. Singly, unassigned, and charge states higher than seven were rejected. Precursor masses previously selected for MS/MS measurement were excluded from further selection for 20 s, and the exclusion mass tolerance was set to 10 ppm. The samples were acquired using internal lock mass calibration on *m/z* 371.1012 and 445.1200. The mass spectrometry proteomics data were handled using the local laboratory information management system (LIMS) (*Türker et al., 2010*).

## Protein identification and label-free protein quantification

The acquired raw MS data were processed by MaxQuant (version 2.0.1.0), followed by protein identification using the integrated Andromeda search engine (*Cox and Mann, 2008*). Spectra were searched against a UniProt *Mus musculus* reference proteome (taxonomy 10090, version from July 9, 2019), concatenated to its reversed decoyed FASTA database and common protein contaminants. Carbamidomethylation of cysteine was set as fixed modification, while methionine oxidation, STY phosphorylation, and N-terminal protein acetylation were set as variable. Enzyme specificity was set to trypsin/P allowing a minimal peptide length of seven amino acids and a maximum of two missed cleavages. The maximum FDR was set to 0.01 for peptides and 0.05 for proteins. Label-free quantification was enabled and a 2 min window for match between runs was applied. In the MaxQuant experimental design template, each file is kept separate in the experimental design to obtain individual quantitative values. Protein fold changes were computed based on Intensity values reported in the proteinGroups.txt file. A set of functions implemented in the R package SRMService (*Wolski et al., 2018*. SRMService – R-Package to Report Quantitative Mass Spectrometry Data; http://github.com/protViz/SRMService; *Wolski et al., 2018*) was used to filter for proteins with two or more peptides allowing for a maximum of three missing values, and to compute p-values using the *t*-test with pooled variance. If all measurements of a protein are missing in one of the conditions, a pseudo-fold change was computed, replacing the missing group average by the mean of the 10% smallest protein intensities in that condition. To determine DARPin and GEPH isoform coverage in the individual pulldown conditions, the data were processed and searched with Proteome Discoverer 2.5 using Sequest and Percolator with Protein Grouping deactivated and only unique peptides were used for quantification.

For relative quantification of gephyrin isoforms, the data were processed with Proteome Discoverer 2.5 and Sequest with variable modifications set to phosphorylation on STY, protein N-terminus acetylation and methionine oxidation, and carbamidomethylation on cysteines as fixed modification. The canonical UniProt mouse database was appended with gephyrin isoform sequences. Only peptides uniquely mapping to gephyrin isoforms C3 (containing insert KHPFYTSPALFMANHGQPIPGLISYSHH ATGSADKR) were used for protein quantification. Based on these settings, isoform C3 was detected with peptides mapping to regions 244–279 and 279–287 and quantified with 8 psms.

## Interactome analysis

Proteins were considered present when detected using at least two unique peptide signatures in all replicates of a given binder. Interactors were considered part of gephyrin complexes when either (1) not present in the control condition or (2) enriched by a log2 fold change in abundance of at least 2.5 in the binder condition with an FDR cut-off of 0.05. These thresholds allowed for complete coverage of known gephyrin interactors. Binders common to multiple interactomes were identified using Microsoft Excel for comparison of ontology and abundances. Venn diagrams were visualized using InteractiVenn (http://www.interactivenn.net/). Protein ontology was identified and grouped, and enrichment determined using WebGestalt over-representation analysis (http://www.webgestalt.org/), Gene Ontology Resource identification (http://geneontology.org/), and UniProt (https://www.uniprot.org/). Interaction networks were generated using STRING version 11.5 and imported to Cystoscape version 3.8.2 for visualization. Network map edges represent putative relationships between protein nodes as identified by STRING. Node size is colored based on functional ontology, and size based on abundance relative to gephyrin in each experiment. Canonical gephyrin interactors include Collybistin (ARGH9), GABA$_A$R subunits (GBRA1, 2,3, GABG2, GBRB2, 3), glycine receptor subunits (GLRB, GLRA), dynein light chain (DYL1, 2), IQSEC3 (IQEC3), Dystrobrevin alpha (DNTA), Ena VASP-like (EVL), MENA (ENAH), the proline *cis-trans* isomerase PIN1, profilins 1 and 2 (PROF1, 2), and neuroligin 2 (NLGN2), reviewed in *Groeneweg et al., 2018*. Protein names used for display are the official UniProt protein ID designation. UniProt protein IDs were used for cross-experiment comparison and ontology searches.

## Statistical tests

Statistical tests and significance are reported in the figure captions. Statistical analysis was performed using Microsoft Excel and GraphPad Prism 8.0. Normality tests were performed on data to evaluate correct application of parametric or nonparametric analysis, with the exception of experiments using small sample sizes (n < 4) where parametric comparisons were used.

## Visual representation

Data plots were generated using Microsoft Excel or GraphPad Prism 8. Images were visualized and processed in Fiji (1.53q). Images brightness was enhanced for display by adjusting the brightness and contrast for display purposes, but when comparing between experimental conditions, all images were enhanced with the same settings to preserve apparent differences in morphology and intensity. Diagrams and figures were arranged in InkScape (version 1.0), and text and tables were arranged using the Microsoft Office Suite. Sequence alignment was performed using ClustalW and visualized using JalView. Heat map generation and hierarchical clustering was performed with Morpheus (https://software.broadinstitute.org/morpheus).

## Materials availability

The use of the anti-gephyrin DARPin constructs presented in this article will be made available following an academic use MTA agreement.

### Project name

Gephyrin interactome from mouse brain lysates using anti-gephyrin antibody and anti-gephyrin DARPins; project accession: PXD033641; project DOI: 10.6019/PXD033641.

## Acknowledgements

We specifically thank Sven Furler, Thomas Reinberg, Joana Marinho, and Jonas Schaefer from the HT-BSF for their assistance in performing the ribosome display DARPin screen. We would like to thank Yuan-Chen Tsai and Marta Figueredo for assistance in the preparation of primary hippocampal neuron cultures. We additionally thank the Functional Genomics Centre Zurich (FGCZ) for carrying out mass-spectrometric analysis and support. We appreciate the help provided by the Protein Production and Purification Core Facility (PTPSP) Lausanne for carrying out medium-scale hFc-tagged DARPin expression. We would also like to thank the members of the Tyagarajan lab for constructive input on manuscript composition.

# Additional information

### Competing interests
Andreas Plückthun: is a cofounder and shareholder of Molecular Partners, who are commercializing the DARPin technology. The other authors declare that no competing interests exist.

### Funding

| Funder | Grant reference number | Author |
|---|---|---|
| Schweizerischer Nationalfonds zur Förderung der Wissenschaftlichen Forschung | 310030_192522 /1 | Shiva K Tyagarajan |
| Schweizerischer Nationalfonds zur Förderung der Wissenschaftlichen Forschung | 310030_192689 | Andreas Plückthun |

The funders had no role in study design, data collection and interpretation, or the decision to submit the work for publication.

### Author contributions
Benjamin FN Campbell, Conceptualization, Data curation, Formal analysis, Methodology, Writing - original draft, Writing – review and editing; Antje Dittmann, Resources, Data curation, Formal analysis, Methodology, Project administration, Writing – review and editing; Birgit Dreier, Resources, Data curation, Formal analysis, Validation, Methodology, Project administration, Writing – review and editing; Andreas Plückthun, Resources, Funding acquisition, Methodology, Writing – review and editing; Shiva K Tyagarajan, Conceptualization, Resources, Supervision, Funding acquisition, Visualization, Methodology, Project administration, Writing – review and editing

### Author ORCIDs
Benjamin FN Campbell (iD) http://orcid.org/0000-0003-4814-2844
Shiva K Tyagarajan (iD) http://orcid.org/0000-0003-0074-1805

### Ethics
The collection of embryonic and adult tissue was performed in accordance with the European Community Council Directives of November 24th 1986 (86/609/EEC). Tissue collection was performed under license ZH011/19 approved by the Cantonal Veterinary office of Zurich.

### Decision letter and Author response
Decision letter https://doi.org/10.7554/eLife.80895.sa1
Author response https://doi.org/10.7554/eLife.80895.sa2

# Additional files

### Supplementary files
• MDAR checklist

• Supplementary file 1. List of DARPin sequences. Amino acid and nucleotide sequences are displayed for all characterized anti-gephyrin DARPins in this study.

### Data availability
All relevant mass spectrometry data has been deposited to the ProteomeXchange Consortium via the PRIDE (https://www.ebi.ac.uk/pride) partner repository. Project Name: Gephyrin interactome from mouse brain lysates using anti-gephyrin antibody and anti-gephyrin DARPins Project accession: PXD033641 Project DOI: https://doi.org/10.6019/PXD033641.

The following dataset was generated:

| Author(s) | Year | Dataset title | Dataset URL | Database and Identifier |
|---|---|---|---|---|
| Campbell BFN, Dittmann A, Dreier B, Plückthun A, Tyagarajan SK | 2022 | Gephyrin interactome from mouse brain lysates using anti-gephyrin antibody and anti-gephyrin DARPins | https://www.ebi.ac.uk/pride/archive/projects/PXD033641 | PRIDE, PXD033641 |

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
