## [Editor Report]

This article describes and validates new tools to study gephyrin biology in the brain, a critical regulator of synaptic inhibition and metabolism. The experiments are compelling, carefully controlled, and lead to a fundamental advance in neuroscience. This article will be of interest to a broad range of neuroscientists including those in synaptic, cellular, and circuit areas.

---

## [Decision Letter]

**Decision letter after peer review:**

Thank you for submitting your article "A DARPin-based molecular toolset to probe gephyrin and inhibitory synapse biology" for consideration by *eLife*. Your article has been reviewed by 2 peer reviewers, and the evaluation has been overseen by a Reviewing Editor and John Huguenard as the Senior Editor. The following individual involved in review of your submission has agreed to reveal their identity: Stephen J Moss (Reviewer #1).

Essential revisions:

1) In experiments aimed to localize or study gephyrin clusters one important requirement is that DARPins do not interfere with the physiological and/or molecular interactions of gephyrins. Are the authors confident that DARPins binding to gephyrin do not alter the gephyrin function/clustering? Along the same line, might the different gephyrin-binding DARPins differentially alter the gephyrin function/clustering?

*Reviewer #1 (Recommendations for the authors):*

The authors are to be commended on the development of novel tools to study the role that gephyrin plays in the formation of inhibitory synapses. The experiments are well controlled and the data presented is of high quality. Moreover, the availability of these new tools will greatly enhance our understanding of the role that gephyrin plays in mediating synaptic inhibition and in regulating the synthesis of Moco and critical enzyme co-factor. The issues that should be resolved before publication are outlined below.

1. Is it possible to confirm the specificity of the DARPs using isoform-specific gephyrin-KO mice?

2. For the mass spec experiments; providing the levels of recovery relative to that seen for gephyrin on a mol/mol basis would aid data interpretation.

3. A number of published studies have suggested that gephyrin interactions with GEPHS for CDC-42 and other small GTPase. Are these proteins recovered in the authors' purifications?

4. KCC2 isoforms are classified as "Phospho-regulators", I believe they would be better classed as "Ion channels and transporters".

*Reviewer #2 (Recommendations for the authors):*

1) In experiments aimed to localize or study gephyrin clusters one important requirement is that DARPins do not interfere with gephyrin physiological molecular interactions. Are the authors confident that DARPins binding to gephyrin do not alter the gephyrin function/clustering? Along the same line, might the different gephyrin-binding DARPins differently alter the gephyrin function/clustering?

2) As mentioned by the Authors anti-gephyrin FingRs are already valid tools (alternative to antibodies) to detect gephyrin clusters in living neurons. The Authors also state that the use of FingRs has been "limited" in its viral delivery in vivo. How can the Authors at this stage exclude that the use of DARPins delivered in vivo will be also limited similarly to FingRs?

3) In keeping with the previous point, FingRs can "self regulate" their expression in order to avoid excessive or aspecific binding. How can the expression of DARPins be regulated? Do the authors think that excessive DARPins expression could lead to artifactual staining?

---

## [Author Response]

Essential revisions:(1) In experiments aimed to localize or study gephyrin clusters one important requirement is that DARPins do not interfere with the physiological and/or molecular interactions of gephyrins. Are the authors confident that DARPins binding to gephyrin do not alter the gephyrin function/clustering? Along the same line, might the different gephyrin-binding DARPins differentially alter the gephyrin function/clustering?

The reviewers raise an important point that should be addressed before using DARPins to label GABA synapses in the functional and intact rodent brain. In the current manuscript, we examined gephyrin clusters in fixed neurons and/or brain tissue, and we do not expect this to bias gephyrin clustering similar to antibody staining.

However, to address the reviewer’s concern we provide new experimental evidence in Figure 3 Supplement 5. We transfected hippocampal neuron cultures with EGFP-tagged gephyrin and probed fixed neurons with either anti-gephyrin DARPin-hFc clones 27B3, 27F3, 27G2, or commercial anti-gephyrin antibody Ab7a. We then quantified the density and size of detected EGFP clusters, finding no difference in the median size of EGFP gephyrin clusters. Therefore, we do not expect DARPins to interfere importantly with morphological detection of gephyrin clustering in fixed neurons.

We are actively collaborating with structural biologists to derive high-resolution structure of DARPin-gephyrin interactions using crystallography/cryo-EM. As these experiments are time intensive this data is beyond the scope of our current manuscript. We anticipate that crystal structure information in tandem with functional analysis of GABAergic inhibition upon loading anti-gephyrin DARPins into neurons for patch clamp electrophysiology, as well as live imaging experiments to track gephyrin cluster size changes after intracellular delivery of anti-gephyrin DARPins would be part of a follow up manuscript.

Reviewer #1 (Recommendations for the authors):The authors are to be commended on the development of novel tools to study the role that gephyrin plays in the formation of inhibitory synapses. The experiments are well controlled and the data presented is of high quality. Moreover, the availability of these new tools will greatly enhance our understanding of the role that gephyrin plays in mediating synaptic inhibition and in regulating the synthesis of Moco and critical enzyme co-factor. The issues that should be resolved before publication are outlined below.1. Is it possible to confirm the specificity of the DARPs using isoform-specific gephyrin-KO mice?

To our knowledge, gephyrin isoform knockout mice do not currently exist to test this. To examine whether DARPin isoform biases described in our manuscript hold true, we reanalyzed our MS/MS data for relative recovery of the C3 gephyrin isoform (which generates peptide fragments that can be uniquely assigned to this variant) compared to total gephyrin levels. As expected, we recovered significantly less of the C3 cassette-containing isoform from precipitations using DARPin 27F3-hFc. This data has been included as a new figure supplement (Figure 6 – Supplement 4).

2. For the mass spec experiments; providing the levels of recovery relative to that seen for gephyrin on a mol/mol basis would aid data interpretation.

Determining the stoichiometry between gephyrin and the different identified interactors would be interesting to calculate for discovering potential roles in postsynaptic apparatus scaffolding and function. For this to be accurate though, we would need additional information about the relative recovery of each type of protein in our preparation. This is because some proteins, for example membrane proteins (such as GABAA and glycine receptors), are expected to be less recovered compared to soluble cytosolic proteins. Indeed, from our dataset (Figure 5 Supplement 2), GABAA receptors are amongst the lower abundance interactors even thought they would be expected to have a higher stoichiometry ratio with gephyrin.

To determine protein complex stoichiometries, the most accurate methods hinge on absolute quantification of peptides from different proteins as in AQUA approaches, which are based on stable-isotope-labeled standards (Gerber SA, Rush J, Stemman O, Kirschner MW, Gygi SP (2003) Proc Natl Acad Sci USA 100(12):6940–6945) or QconCAT (Beynon RJ, Doherty MK, Pratt JM, Gaskell SJ (2005) Nat Methods 2(8):587–589). We did not include such standards as this would have involved a significantly longer method development phase. However, we agree that label-free methods using normalized spectral counting (e.g. protein abundance index PAI, APEX) or signal intensity (e.g. top3) would also be an option to estimate relative protein amounts, but only an estimate at best. Spectral counting-based methods suffer from narrower dynamic range limits and under sampling, while intensity-based methods hinge on the reproducible identification of the same peptides across all samples, which is not always the case and may lead to incomplete quantification matrices.

Therefore, we suggest that it may be misleading to represent mol:mol relationships based on our dataset, though the identified interactors could be specifically interrogated for their stoichiometric relationship with gephyrin on an individual protein basis.

3. A number of published studies have suggested that gephyrin interactions with GEPHS for CDC-42 and other small GTPase. Are these proteins recovered in the authors' purifications?

We did not detect CDC42 as a gephyrin interactor after applying thresholding. The Rho GTPase activating protein 32 (RGH32, gene name Arhgap32), the Rho GTPase activating protein 44 (RGH44, gene name Arhgap44), and ArfGAP with GTPase domain were detected in all DARPin-determined gephyrin interactomes. RGH32 protein was also previously identified as a gephyrin interactor using proximity ligation proteomics (DOI: 10.1126/science.aag0821).

4. KCC2 isoforms are classified as "Phospho-regulators", I believe they would be better classed as "Ion channels and transporters".

In this manuscript we used the protein names *en lieu* of gene name identifiers, though this could lead to confusion in some cases. KCC2A is the protein name for the kinase CaMKIIα (https://www.uniprot.org/uniprotkb/P11798/entry). The Uniprot ID, gene names, and common names for all interactors are listed in the file “Figure 5 – Source Data 1”. As the KCC2 identifier is particularly confusing, we have added this clarification to the discussion where CaMKIIα is mentioned.

Reviewer #2 (Recommendations for the authors):(1) In experiments aimed to localize or study gephyrin clusters one important requirement is that DARPins do not interfere with gephyrin physiological molecular interactions. Are the authors confident that DARPins binding to gephyrin do not alter the gephyrin function/clustering? Along the same line, might the different gephyrin-binding DARPins differently alter the gephyrin function/clustering?

This recommendation is addressed in the essential revisions section above.

(2) As mentioned by the Authors anti-gephyrin FingRs are already valid tools (alternative to antibodies) to detect gephyrin clusters in living neurons. The Authors also state that the use of FingRs has been "limited" in its viral delivery in vivo. How can the Authors at this stage exclude that the use of DARPins delivered in vivo will be also limited similarly to FingRs?

We erred in our discussion of this point. Our statement stemmed from the potential issue of FingR expression using fluorescently-tagged FingRs’ transcriptional control system where nuclear accumulation of fluorescent signal could interfere with fluorescence-based detection of perisomatic gephyrin clusters. Of course, specifically relating this to FingRs was incorrect, as this issue would exist if DARPins were expressed in neurons using the same expression titration system as well. We have now corrected this in the discussion, and thank the reviewer for pointing out this error.

(3) In keeping with the previous point, FingRs can "self regulate" their expression in order to avoid excessive or aspecific binding. How can the expression of DARPins be regulated? Do the authors think that excessive DARPins expression could lead to artifactual staining?

We did not assess DARPins for this application in this manuscript, though we intend to explore the use of DARPins as in vivo tools in future studies. The similar genetic size of DARPins to FingRs allows them to be used in similar expression control systems. The stability of DARPins is expected to require their expression to be controlled through either a transcriptional feedback system, or inducible methods. Otherwise DARPins would be expected to accumulate intracellularly (which could interfere with protein homeostatic processes, as well as gephyrin function potentially).